# The speleothem oxygen record – a proxy for thermal or moisture changes? A case study of multiproxy records from MIS 5/MIS 6 speleothems from the Demänová Cave System.

Jacek Pawlak[1]

[1]Institute of Geological Sciences, Polish Academy of Sciences, Warsaw, Poland 00- 818

*Correspondence to*: Jacek Pawlak (dzeq@twarda.pan.pl)

Abstract. Speleothems are an important source of paleoclimatic information for the terrestrial environment. The

basic advantages of speleothems are their high preservation potential, the possibility of precise dating by the U-series method and many different proxies, such as stable isotopes, trace elements, and microfabrics, which can be interpreted in terms of paleoclimatic conditions. Currently, central Europe is located in a transitional climate zone under the influence of both oceanic and continental climates. However, in the past, the region could have been under a stronger continental climate influence during cold glacial episodes or a stronger oceanic climate

influence during wetter interglacial episodes. Long-term speleothem records can add new beneficial data about past climate changes in the region. The multiproxy record of JS9 stalagmite, collected in the Demänová Cave System (Slovakia), represents a ca. 60 ka period (143 – 83 ka). A multiproxy interpretation of the JS9 record shows that long-term $\delta^{18}$O trends can be interpreted as global/regional temperature changes, while short-term $\delta^{18}$O signals reflect changes in humidity. In contrast to the records from the Alps and the northern Tatra

Mountains, the $\delta^{18}$O record of speleothem JS9 shows instantaneous decreasing episodes during termination II. This shows that the Carpathian Belt was an important climatic barrier at that time.

## 1. Introduction

Speleothems are important paleoenvironmental archives (Lachniet, 2009; Fairchild and Treble, 2009; Fairchild and Baker, 2012; Koltai et al., 2018; Kern et al., 2019). Many different paleoclimatic proxies are currently studied

in speleothems, such as stable isotopic compositions, trace element contents, calcite microfabrics, and isotopic compositions of water from speleothem inclusions (Fairchild and Treble, 2009; Wong and Breecker, 2015; Frisia, 2015; Demény et al., 2017; Baker et al., 2019). Nevertheless, the $\delta^{18}$O proxy is still the most commonly used for regional and global comparisons (Lisiecki and Raymo, 2005; McDermott et al., 2011; Govin et al., 2015). Therefore, understanding which climatic factor has the strongest influence on $\delta^{18}$O composition in a given site is

a crucial problem (Lachniet, 2009).

Basically, the $\delta^{18}$O value of speleothems reflects the oxygen isotopic composition of precipitation. The isotopic composition of precipitation depends on global factors, such as the mean isotopic composition of ocean surface water. On a long-term scale, the global volume of glacial ice has an impact on the $\delta^{18}$O value of ocean surface water (Dansgaard, 1964). Currently, the Atlantic Ocean is the main source of vapor for precipitation in Europe.

Other potential sources are the Mediterranean Sea, Black Sea, and Nordic Seas (Ionita, 2014). Water from the Mediterranean Sea surface is enriched in $^{18}$O in comparison to water from the Atlantic Ocean. During glaciations,

the Fennoscandian ice sheets (which is enriched in [16]O) influence the atmospheric circulation in Central Europe and consequently influence the isotopic composition of meteoric water (Bianchi and McCave 1999; Elmore et al. 2015). During deglaciations, cold melting waters can slow down the circulation of Atlantic currents. The consequence of such a situation for Central Europe may be the limited influence of the Atlantic Ocean and the stronger influence of enriched [18]O moisture transported from the Black Sea and Mediterranean region (Celle-Jeanton et al., 2001). The other regional factor shaping $\delta^{18}O$ in Europe is the continental effect (McDermott et al., 2011). Finally, the isotopic composition of rainwater is modified at the precipitation site by local factors, such as the altitude, amount effect, and local temperature (Drysdale et al., 2005; Moseley et al., 2015).

The isotopic composition of dripping waters can also be modified inside the soil and in the epikarst zone by evaporation and priori calcite precipitation (PCP; Baker et al., 2019). However, PCP affects more the $\delta^{13}C$ value. In a cave environment, the isotopic fractionation between the dripping water and crystallizing calcite depends on the cave air temperature. The cave air temperature usually reflects the mean annual temperature in the near-cave area. Additionally, the isotopic composition of crystallizing calcite can be modified by kinetic effects if the relative humidity of the cave air is below 100% (Dorale and Liu 2009).

Recently, more than one dozen speleothem records of the last interglacial age have been obtained from the European continent ( Meyer et al., 2008; Couchoud et al., 2009; Moseley et al., 2015; Vansteenberge et al., 2016; Demény et al., 2017; Pawlak et al., 2019; Pawlak et al., 2020). The temperature, amount of precipitation at the cave site, and changes in the main source of vapor for precipitation are considered the main factors driving the $\delta^{18}O$ value of precipitated calcite. However, for records from the Alps and Central Europe, temperature seems to be a more important factor (Moseley et al., 2015; Kern et al. 2019; Pawlak et al., 2019; Comas-Bru et al., 2020; Pawlak et al., 2020). This result is in accordance with the observations made by Różański et al. (1993), which show that the isotopic composition of rainfall in temperate regions of Europe mostly depends on the local temperature. However, distinguishing which factor is the most dominant is not always a simple task. For example, coastal sites are more influenced by the amount of precipitation (Couchoud et al., 2009; Vansteenberge et al., 2016).

In contrast to most European records, the records from the Middle East seem to be influenced by more factors, such as the amount of precipitation, temperature, and changes in the main source of vapor for precipitation (the source effect; Bar-Matthews et al., 2003; Nehme et al., 2015). These factors can be linked to changes in the prevailing circulation patterns, the impact of evaporation on the Mediterranean Sea surface $\delta^{18}O$ and the lower amplitude of long-term mean annual temperature changes during the last interglacial period at lower latitudes (Rybak et al. 2018).

Currently, Slovakia is influenced by two main types of climates: a boreal and fully humid climate with warm summers (Dfb) in the east and a warm temperate and fully humid climate (Cfb) in the west (Kottek et al. 2006). However, in the past, the local climate could have been more continental during colder and drier glacial periods and more transitional during warmer interglacial periods (Feurdean et al. 2014). The new long-term speleothem records can add beneficial data about past climate changes in this region. This paper presents an ca. 60 ka long multiproxy record ($\delta^{18}O$, $\delta^{13}C$, Mg, Sr, Ba, Na, P, Fe, Mn, and Si) of the MIS 5/MIS 6 stalagmite collected in the Demänová Cave System, which is located in Slovakia. The interpretation of these proxies is focused on

distinguishing dry continental climate phases from more wet transitional climate episodes. Additionally, the interpretation of stable isotopic compositions and trace element contents helps to distinguish which factor had the strongest influence on the shape of the $\delta^{18}O$ record: the local temperature, humidity, or source effect.

## 2 Study Settings

The Demänová Cave System (DCS) is located in the Low Tatra Mountains (Fig. 1 A), Western Carpathians,
Slovakia (19.58°E; 49.00°N, 837 m.a.s.l.). The DCS is 41.4 km (Fig. 1 B) long, and its denivelation is 196 m (Herich, 2017). The DSC includes ten caves connected to each other. Sample JS9 was collected in Demänovská Slobody Cave (Fig. 1 B). The DCS has nine cave levels that can be correlated with fluvial terraces of the Demänovka Stream according to Droppa (1966, 1972). The DCS was generated by fluvial processes. The cave corridors were formed by allochthonous sinking streams during the late Pliocene (Bella, 1993). The host rocks of
the DCS are Middle Triassic carbonate rocks, primarily the Gutenstein and Annaberg limestones (early Anisian), organodetritic limestones (Late Anisian), and Ramsau dolomite (Ladinian) (Droppa, 1957; Gaál, 2016; Gaál and Michalík, 2017).

The DCS is located in a transitional climate zone between oceanic and continental climates (Sotak and Borsanyi, 2004; Kottek et al. 2006). There are two meteorological stations located near the DSC: the first is located in
Liptovský Mikuláš town close to the Demänovská valley entrance (49.07°N 19.61°E, 570 m.a.s.l.), and the second is located at Chopok peak under the influence of a cold mountain climate (48.94°N, 19.59°E, 2008 m.a.s.l.). Chopok peak has a colder and wetter climate (mean annual temperature of -0.1°C; mean annual precipitation of 1325 mm), while Liptovský Mikuláš town has a warmer and drier climate (mean annual temperature of +6.9°C; mean annual precipitation of 537 mm). The local climate has strong seasonality; the coldest and driest months are
January and February, while the warmest are July and August, and the greatest amount of precipitation is usually noted in June and July. Despite the high altitude and thermal gradient along the valley, the mean annual $\delta^{18}O$ value of precipitation at Chopok peak (-10.43 ‰) is similar to the value at Liptovský Mikuláš (-10.92 ‰) (Holko et al., 2012). The average cave temperature measured in 2015-2016 at several sites in the DSC was 6.3 ± 0.6°C (Hercman et al. 2020).

Based on a large population of U-series ages, several stages of speleothem crystallization, which developed chiefly in the warmer periods of the Pleistocene and in the Holocene, were distinguished in the DCS (e.g., Hercman et al., 1997; Hercman, 2000; Hercman et al., 2020; Bella et al., 2021). Field observations along with a chemical study of underground water in caves (Motyka et al., 2005) suggest that many speleothems are still growing in the DCS. Sample JS9 was collected in the northern part of Demänovská Slobody Cave (Fig. 1 B).

## 3 Methods

### 3.1 Petrography

The whole JS9 stalagmite profile was analysed by a Nikon Eclipse LV100POL microscope, including microfabric structures, the appearance of calcite crystals, potential discontinuities and porosity. The analysis of speleothem microfabrics and microfabric log construction was based on the methodology proposed by Frisia (2015). The
microscopic analyses were performed at the Institute of Geological Sciences at the Polish Academy of Sciences (Warsaw, Poland).

**3.2 U-series and age-depth model**

Ten calcite samples (0.1 – 0.5 g) were collected by drilling the JS9 speleothem through its growing axes. The samples were drilled as thin as possible with an average thickness of 2.5±0.2 mm. The chemical preparation of the samples was performed at the U-series Laboratory of the Institute of Geological Sciences, Polish Academy of Sciences (Warsaw, Poland). To control the efficiency of the chemical procedure, at its beginning, a spike ($^{233}$U, $^{236}$U and $^{229}$Th mixture) was added to the samples. In the first step of the chemical procedure, the samples were heated to decompose potential organic matter. Then, the samples were dissolved in nitric acid. Finally, uranium and thorium were separated from the solution by chromatographic methods using TRU Resin (Hellstrom, 2003). In addition to the regular samples, the internal standards and blank samples were processed by the same procedure. U and Th isotopic compositions of all the samples and standards were measured at the Institute of Geology of the CAS, v. v. i. (Prague, Czech Republic) by a double-focusing sector-field ICP mass analyser (Element 2, Thermo Finnigan MAT). The spectrometer settings were at a low mass resolution ($m/\Delta m \geq 300$). The obtained measurements were corrected for background counts and chemical blanks. The final results were reported as the activity ratios (Table 1). The final U-series ages were calculated by taking into account the newest decay constants (in $yr^{-1}$): $\lambda_{238} = (1.55125 \pm 0.0017) \cdot 10^{-10}$ (Jaffey et al., 1971), $\lambda_{234} = (2.826 \pm 0.0056) \cdot 10^{-6}$ (Cheng et al., 2013), $\lambda_{232} = (4.95 \pm 0.035) \cdot 10^{-11}$ (Holden, 1990) and $\lambda_{230} = (9.1577 \pm 0.028) \cdot 10^{-6}$ (Cheng et al., 2013). The reported age errors were estimated by using error propagation rules. All measurement errors were considered except for the decay constant.

A modified version of the Hellstrom algorithm was applied for the correction of obtained ages (Hellstrom, 2006), with the assumption of initial contamination by $^{230}$Th, $^{234}$U and $^{238}$U isotopes. The algorithm searches for the lowest values of initial contamination by $^{230}$Th, $^{234}$U and $^{238}$U isotopes from detrital sources, which were able to correct series of ages in stratigraphic order (Błaszczyk et al. 2021). The age-depth model was calculated by the MOD-AGE algorithm (Hercman and Pawlak, 2012).

**3.3 Stable isotopes**

The samples for measurement of stable isotopic composition were drilled by a Micro-Mill with a drill bit diameter of 0.1 mm. The final number of obtained samples was 290. In the first stage, the JS9 speleothem was sampled along its growth axis at a resolution of one sample/mm. The lower part of the stalagmite (0 to 40 mm) was also sampled at a resolution of one sample/0.3 mm to minimize the difference in resolution between the lower and upper parts of the record that was caused by growth rate differences. The isotopic compositions of O and C were measured by a Thermo Kiel IV carbonate device connected to a Finnigan Delta Plus IRMS spectrometer in dual inlet mode. The results were normalized to three international standards, NBS 19, NBS 18, and IAEA CO 8, and were reported relative to the V-PDB international standard. The analytical precision (1σ) was better than 0.03 ‰ and 0.08 ‰ for $\delta^{13}$C and $\delta^{18}$O, respectively. The reproducibility was checked by measurement of two internal

standards after every 12 samples (for δ13C: ±0.03 ‰; for $\delta^{18}O$: ±0.08 ‰). The analyses were performed in the Stable Isotope Laboratory (Institute of Geological Sciences, Polish Academy of Sciences) in Warsaw.

**3.4 Trace elements**

The trace element content was analysed from thin sections by an Analyte Excite Excimer Laser Ablation System with a wavelength of 193 nm connected to an Element 2 inductively coupled plasma mass spectrometer (Thermo Finnigan); using a laser output of 50% with 10-Hz pulses, a fluence of 2.44 $J/cm^2$ was achieved. The width of each line was 50 μm, and the laser speed during each scan was 5 μm/s. Additional details of the LA-ICP-MS analytical procedure followed the procedure described by Eggins et al. (1997). Measurements of near-surface trace element contents, namely, Mg, Sr, Na, Ba, P Si, Fe, and Mn, were performed at medium resolution. The obtained raw data were normalized to Ca. Finally, the data were smoothed by the adjected averaging method using 10 nearby data points.

**4 Results**

**4.1 Petrography**

The results of petrographic studies are presented in Fig. 2. The JS9 sample is a 155 mm long columnar stalagmite with a diameter of 80 mm. Macroscopically, JS9 stalagmite is constructed from laminated calcite (Fig. 2 A). The colour of the laminae changes from light crème to dark brown (Fig. 2 A). Between 75 and 85 mm, the stalagmite has a grey colour. The light crème laminae between 40 and 75 mm have a zone of macroscopically visible porosity in the axial part of the stalagmite. A microscopic analysis of calcite crystal appearances and the identification of textural features show that most of the observed stalagmite is composed of columnar polycrystals with length to width ratios that are usually > 10:1. "Fibre-like" calcite individuals compose each polycrystal. The overall appearance of this layer is similar to spherulite, which consists of bundles of elongated crystals bending outward (Fig. 2 B, C). The polycrystals show brush extinction that converges away from the substrate when the rotating table is turned clockwise. The characteristics mentioned above are similar to those described by Frisia (2015); these characteristics indicate that the columnar radiaxial fibrous (Crf) texture is a dominant fabric in the JS9 stalagmite (Fig. 2 B). Parts of stalagmites containing Crf fabric are usually separated by thin layers dark in cross-polarized light consisting of small calcite crystals and detrital material (Fig. 2 C). The appearance of these thin layers indicates the micrite (M) fabric described by Frisia (2015). Micrite fabric is most common in the middle part and in the youngest layers of JS9 stalagmite (Fig. 2 C).

**4.2 U-series dating and age-depth model construction**

The results of 10 U-series dates are presented in Table 1. The reported errors are 2σ, and they vary from 0.8 to 2.8%. The analysed samples do not show any visible detrital contamination at the dissolution stage. However, 4 of the measured samples have a $^{230}Th/^{232}Th$ ratio lower than 300. In the case of measurement by mass spectrometry, these samples should be considered contaminated by detrital admixtures (Hellstrom, 2006). Therefore, the whole profile was corrected by using a modified procedure proposed by Hellstrom (2006). The result from correction shows that the corrected ages are within the error range of the uncorrected ages (Table 1).

Based on the U-series dating results, age-depth models for the JS9 stalagmite (Fig. 3) were created. According to the obtained age-depth model, deposition of the JS9 stalagmite started at 142±4 ka and ended at 84±3 ka. The growth rate of JS9 stalagmite was not uniform. From 142 to 112 ka, its growth rate was relatively slow at 1.4 mm/ka; after 112 ka, it rapidly grew at a rate of 11.5 mm/ka, which ended at 109 ka, and then the growth rate slowed to 1.9 mm/ka. The last intensive change in the growth rate occurred after 95 ka, when it increased to 4.2 mm/ka.

**4.3 Stable Isotopes**

The obtained isotopic records (Fig. 4 A, B) cover the interval from the late MIS 6 to MIS 5a. The mean value of the $\delta^{18}O$ record is -7.05‰ (Fig. 4 A), and its value varies in a range from -9‰ to -5.7‰. The $\delta^{18}O$ signal expresses short time changes, and the average value of its amplitude is ca. 0.8‰ (Fig. 4 A). At the end of MIS 6, an increase in $\delta^{18}O$ values is interrupted by a 1.2‰ instant decline. The next important change occurs at the boundary between MIS 5d and MIS 5c – an episode of elevated $\delta^{18}O$ values greater than -6‰. Finally, at the boundary between MIS 5c and MIS 5b, a short episode occurs, during which $\delta^{18}O$ values decrease to -9‰ (Fig. 4 A). The $\delta^{13}C$ record expresses changes in its values from -1‰ to -9.8‰ (Fig. 4 B). The average amplitude of $\delta^{13}C$ values for changes over a short time period is close to 1‰. In contrast to the $\delta^{18}O$ record, the $\delta^{13}C$ curve is dominated by episodes of lower and higher values. They are divided by large-scale shifts (Fig. 4 B). From 143 to 139 ka, $\delta^{13}C$ values rise to -1‰. From 139 to 130 ka (an interval of 9 ka), $\delta^{13}C$ values decrease from -2 to -7‰ (5‰), and the values are low ca. -8‰ until 110 ka. From 110 to 107 ka, $\delta^{13}C$ values increase from -9.3 to -2.6‰ (6.7‰) and decrease from -1 to -8.5‰ (7‰) at 101 ka. From 100 to 85 ka, $\delta^{13}C$ values oscillate at approximately -8.2‰. After 85 ka, $\delta^{13}C$ values increase to -5.6‰ (Fig. 4 B).

**4.4 Trace elements**

Trace element content results are presented in Fig. 4 C - J. Mg, Sr and Ba contents do not show clear correlations or anticorrelations (Fig. 4 C, D, E). However, few single Mg extremes are clearly in phase or out of phase with the Sr content. For example, at 138 ka, they are in phase, and at 122.5 ka, they are out of phase (Fig. 4 C, D). Generally, Mg and Ba records are similar from 138 to 101 ka, while Sr and Ba contents show more similarities before 138 ka and after 101 ka. Mg and Sr contents have minimums at 101 ka, while the minimum Ba content occurs at 93 ka and is mimicked by a decreasing Sr content during the same period.

Na, P, Fe, and Mn contents show a similar pattern (Fig. 4 F, G, H, I), and they have three intervals of increased values: before 138 ka, from 106 to 98 ka, and after 92.5 ka. From 98 to 92 ka, Na, P, Fe, and Mn contents are lower. This trend is also similar for Ba and Sr contents, and only the Mg content has a different pattern in this interval. The Fe content has the largest number of peaks (Fig. 4 H), several of which are also shown by Na, P, and Mn contents (Fig. 4 F, G, I). In comparison to those four records, the P content has the lowest peak amplitude (Fig. 4 G).

The Si content shows a few different patterns. The most visible pattern is a short maximum at 102 ka, which transitions into a minimum almost immediately at 101 ka, but the amplitude of changes in the Si content is

relatively low. However, from 122 to 102 ka, it has an increasing trend, and a similar trend can be observed for
Na and P contents (Fig. 4 F, G, J). Similar to Fe, Mn, P, Na, and Ba contents, the Si content has higher values
before 138 ka and after 98 ka.

## 5. Discussion

### 5.1 Drivers of $\delta^{18}$O in DCS speleothems

The meteorological data collected in Slovakia show that there is a significant relation between the $\delta^{18}$O value of
atmospheric precipitation and the mean annual temperature ($R^2 = 0.73$; Holko et al 2012). The influence of the
annual amount of precipitation on the $\delta^{18}$O value is less obvious ($R^2 = 0.48$; Holko et al 2012). In the DCS
region, the temperature gradient for the $\delta^{18}$O of precipitation is 0.36 ‰/°C (Holko et al 2012). The current $\delta^{18}$O
precipitation value changes during the year from -16‰ in February to -6‰ in July (Holko et al 2012). Currently,
nearly 65% of precipitation occurs during the spring and summer months (April – September). Therefore, the
seepage water is biased by the seasonal effect, and its mean $\delta^{18}$O value is higher than expected based on the
mean annual temperature.

Recently, a meta-analysis of cave drip water and precipitation monitoring records showed that in climates with
mean annual temperatures lower than +15 C° and aridity indices higher than 0.65, the isotopic composition of
dripping water is not affected by evaporation (Baker et al. 2019). The current climate of the study region fits
these conditions of a mean annual temperature lower than +15°C and an aridity index higher than 0.65.
Therefore, current interglacial conditions are more conducive for $\delta^{18}$O signals to reflect the regional temperature
conditions and isotopic composition of meteoric water. In contrast, during glacial episodes, when the local
climate was more continental and the aridity index was lower, theoretically the $\delta^{18}$O record in this study could
have been affected by the evaporation effect. However, due to lower mean annual temperature the effect should
not be high.

The main factor shaping the $\delta^{18}$O value of western and central European speleothems during the Holocene and last
interglacial period was temperature (Moseley et al., 2015; Kern et al. 2019; Comas-Bru et al., 2020; Hercman et
al., 2020). For example, stalagmites from Cobre cave, which is located on the northern coast of Spain (Fig. 5),
reflect changes in the oceanic moisture isotopic composition, which is dependent on temperature (Rossi et al. 2014;
Fig. 6). Similarly, the $\delta^{18}$O record from Han-sur-Lesse Cave in Belgium (Fig. 6) is driven by temperature and by
changes in the isotopic composition of the ocean surface (Vansteenberge et al., 2016). The main trend of the $\delta^{18}$O
record from the Hungarian stalagmite collected in Baradla Cave (Fig. 5), reflects temperature changes (Demény
2017).

According to all the facts listed above it can be assumed that, in the long time scale, the temperature effect on
atmospheric precipitation should be the main factor shaping the $\delta^{18}$O value of the DSC stalagmites. The whole
temperature effect on $\delta^{18}$O value of speleothem calcite is lover than presently observed $\delta^{18}$O temperature gradient
of precipitation (0.36 ‰/°C; Holko et al 2012) due to the opposite temperature effect on calcite crystallisation (-
0.18 ‰/°C). Therefore, the expected net effect should be ca 0.18 ‰/°C. However, in the short time scale the main
temperature effect can be changed by humidity/aridity effects like it was described in case of Bourgeois–Delaunay
Cave (Couchoud et al., 2009).


## 5.2 Drivers of $\delta^{13}C$ and trace elements in DCS speleothems

The $\delta^{13}C$ value of speleothem calcite depends on the proportion of $CO_2$ from a soil source and from a host rock source. $CO_2$ from a soil source can reflect the changes in the isotopic composition of atmospheric $CO_2$ and

additionally is enriched in $^{12}C$ due to biological activity. Due to that fact a well-developed soil cover results in a lower $\delta^{13}C$ value. The level of soil development depends on climatic conditions such as temperature and humidity. Currently, the vegetation cover over the DCS is dominated by mixed forests, including mountain-type forests and grasslands connected with mountain slope activity (Hercman et al., 2020). During the cold episodes of the last glacial period, the Slovakian landscape was dominated by boreal trees, tundra dwarf shrubs and grasslands

(Feurdean et al., 2014; Jankovska et al., 2002). The beginning of the Holocene was marked by the development of temperate-type forests (Feurdean et al. 2014). Recently, a set of five Holocene speleothems from different parts of the DCS was investigated (Hercman et al., 2020). The mean $\delta^{13}C$ value varied from ca. -8 ‰ to ca. -2 ‰ and was site dependent (Hercman et al., 2020). Despite these values, the shape of all Holocene $\delta^{13}C$ records from the DCS reflects the changes from boreal and tundra vegetation types to temperate and mountain forest types. This change

suggests that the $\delta^{13}C$ proxy in the DCS can be interpreted in terms of vegetation changes.

Trace elements such as Mg, Sr and Ba are transported in water solution. Their relative abundance depends on the time of water residence and on the host rock composition (Fairchild and Treble, 2009). Drier conditions result in longer water residence times. The process of dolomite rock dissolution is slower than the process of limestone

rock dissolution. Therefore, during longer water residence times, the contribution of trace elements from dolomite host rock sources increases. Dolomite normally contains less Sr and Ba than calcite, resulting in higher Mg/Ca and lower Sr/Ca and Ba/Ca ratios during drier conditions (Roberts et al., 1998; Tremaine and Froelich 2013; Rossi et al., 2014). The DCS is developed mostly in Gutenstein limestones and Ramsau dolomites (Gaál, 2016). Therefore, the dissolution of both limestones and dolomites is possible in the DCS. On the other hand, prior calcite

precipitation (PCP) can also occur during dry episodes when the water residence time is longer. However, the PCP results in an increase in all X/Ca ratios because $Ca^{2+}$ cations are preferred during calcite crystallization (Tremaine and Froelich 2013). Episodes of synchronous increases in Mg/Ca and Sr/Ca ratios are not observed in the record in this study.

Elements, such as Fe, Mn, and Si, may be transported as detrital particles or submicron-size colloids (Fairchild

and Treble, 2009). Additionally, all elements that can be incorporated into the calcite structure can be transported as absorbed ions on the clay mineral structure. During drier periods under higher aeolian supply conditions, particles can be transported into the cave environment without water transportation (Hu et al. 2005).

## 5.3 Temporal evolution of environmental proxies in the JS9 speleothem


Recently, a set of five speleothems of Holocene age from the DCS was investigated (Hercman et al. 2020). Holocene $\delta^{18}O$ records from DCS speleothems reflect the same pattern (Hercman et al. 2020). Additionally, the mean value for all these records is similar. This proves that the $\delta^{18}O$ value from DCS speleothems reflects the final regional climatic conditions. Comparing Holocene $\delta^{18}O$ values with the record in this study can be useful

for its interpretation. Ninety-five percent of Holocene $\delta^{18}O$ values from the DCS are in the range from -7.6% to -6.8% with a mean value of -7.2‰ (Fig. 4 A). There are five periods in which the $\delta^{18}O$ value of the JS9 stalagmite was slightly different than the $\delta^{18}O$ mean value of the Holocene: 143 – 135 ka; 127 – 123 ka; 116 – 113 ka; 108 – 101 ka; and 94 – 92 ka (Fig. 4 A).

The $\delta^{13}C$ and $\delta^{18}O$ values in the JS9 stalagmite change dynamically during the 143 – 130 ka period. The short episode of elevated $\delta^{18}O$ and $\delta^{13}C$ values from 143 to 137 may be the result of drier conditions with a higher level of evaporation at the end of MIS 6 (Gascoyne, 1992; Genty et al., 2006, Couchoud et al., 2009). During the 143 – 137 ka time interval, Mg and Ba contents are elevated, while the Sr content is lower (Fig. 4 C, D, E). Trace element contents (Mg, Ba, and Sr) support this interpretation. From 143 to 137 ka, the Fe, Mn and Si contents

are elevated. This result may be related to the lack of developed soil cover and increased frost erosion of the surface above the cave. The other explanation of elevated values of listed proxies is the presence of micrite (Fig 4 K). However, this explanation also link to dryer conditions (Frisia, 2015). New data revealing $CO_2$ concentration changes during MIS 6 (Shin et al. 2020) show a local minimum $CO_2$ concentration between 142 and 138 ka. Therefore, the episode recorded in the JS9 stalagmite may be global in nature.

After 137 ka, the change in the negative value observed in the $\delta^{13}C$ record (ca. 5‰) is ca. two times larger than that observed in the DCS $\delta^{13}C$ records at the beginning of the Holocene (ca. 2.5 ‰; Hercman et al., 2020). This may reflect the more substantial change in the environment from periglacial tundra conditions to temperate forests. According to data from other speleothems, the long-term tendencies for $\delta^{13}C$ and $\delta^{18}O$ are clearly related to improvements in thermal conditions after the MIS 6 glacial maximum and before the MIS 6/MIS 5e transition

(Pawlak et al., 2020; Moseley et al., 2015; Meyer et al., 2008; Holzkamper et al., 2004).

Termination II (T II) in the JS9 stalagmite is highlighted by rapid decrease of ca. 1.2‰ in the $\delta^{18}O$ value (Fig. 4 A). This decrease is in contrast to the $\delta^{18}O$ signal recorded in speleothems from the northern rim of the European Alps (NALPS; Fig. 6). The observed positive shift in Alpen $\delta^{18}O$ records during termination II is the result of two

processes: the improvement of thermal conditions and the change from winter-dominated precipitation to summer-dominated precipitation (Moseley et al. 2015; Meyer et al., 2008; Holzkamper et al., 2004; Fig. 6). Similarly, in the caves on the northern slopes of the Tatra Mountains, Termination II is visible as a positive change in $\delta^{18}O$ (Pawlak et al., 2020). However, in the case of the $\delta^{18}O$ record from Magurska Cave (Tatra Mts., Poland), the change towards positive values is preceded by an instant decrease of ca. 2‰ in its value (Fig. 6). The difference

between the Low Tatra Mountains and caves located on the northern slopes of the Tatra Mountains (ca. 39 km towards the NE) is that the Tatra Mountains were an important climatic barrier for moisture at that time. In the case of the JS9 stalagmite, the $\delta^{13}C$ and trace element contents did not show any signal, which could be equivalent to the rapid 1.2 ‰ negative change in the $\delta^{18}O$ record at 130 ka (Fig. 4). Therefore, the recorded 1.2 ‰ negative shift must be caused by factors that affect only the $\delta^{18}O$ proxy. Additionally, the $\delta^{18}O$ value after T II remains at

the lower level until the end of MIS 5e. Its average value for MIS 5e (-7.6 ‰) is ca. 0.4 ‰ lower than the average $\delta^{18}O$ value (-7.2 ‰) for Holocene speleothems of the DCS (Hercman et al. 2020). According to the present temperature gradient in Slovakia, the -0.4 ‰ difference could be interpreted as a lower mean temperature of MIS 5e ca. 1C° in comparison to the Holocene. However, this simple interpretation of has a low probability. Other reasons, such as changes from summer-dominated precipitation to winter-dominated precipitation or more humid

than conditions at present, are more probable. Generally, in Central Europe, the beginning of MIS 5e is related to changes from continental to more transient climates (Demény et al., 2017; Moseley et al., 2015). Therefore, the other reason for the observed shift may be the source effect and rapid increase in the depletion of $^{18}$O moisture from the Atlantic source. The negative shift in the $\delta^{18}$O record observed in JS9 speleothems is similar to the change observed in Mediterranean records (Antro del Corchia, Soreq, Peqiin; Fig. 6). This type of shift in Mediterranean

records can be explained as a source effect, when the change in the $\delta^{18}$O composition of speleothems is caused by the change in the $\delta^{18}$O value of the Mediterranean Sea surface, which is documented by marine cores, and the change in the proportion of moisture from Atlantic and Mediterranean sources (Bar-Matthews et al., 2003; Nehme et al., 2015). These are similar to records from Antro del Corchia (Drysdale et al., 2005; CC5 stalagmite), which has an approximately 2.5 ‰ decrease in its $\delta^{18}$O values during termination II at ca. 130 ka (Fig. 6). In the case of

the JS9 stalagmite, the observed 1.2 ‰ negative change is the result of a muted response. This response must have been caused by local or regional effects, which were stronger than the thermal effect at that time. The possible effect that may have caused the lower $\delta^{18}$O value is the circulation effect and changes from sources of precipitation, such as the Adriatic Sea or Black Sea, to Atlantic sources and vapor recycling over the European continent (Drysdale et al 2005). The instant change to negative values may have been caused by source and continental

effects, which overcame the temperature effect at that time.

The MIS 5e interval in the JS9 stalagmite can be divided into two parts. The first part (127 – 123 ka) has lower values of $\delta^{18}$O and $\delta^{13}$C (Fig. 4 A, B). The Mg content during this period is lower than its average value, with a local minimum at 124 ka (Fig. 4 C). Usually, low $\delta^{13}$C and $\delta^{18}$O values can be interpreted as a sign of wetter and

colder conditions (Gascoyne, 1992; Genty et al., 2006; Couchoud et al., 2009). However, at the global scale, the 127 – 123 ka period is a time of the highest sea level and warm temperature conditions (Goelzer et al., 2016). At that time, other European records show high sensitivity to changes in the amount of precipitation. For example, the $\delta^{18}$O record of a stalagmite from Bourgeois–Delaunay Cave (Couchoud et al., 2009) shows millennial variability with amplitudes lower than 1‰ (Fig. 6). The $\delta^{18}$O changes are reiterated by $\delta^{13}$C changes (Couchoud

et al., 2009). The authors' interpretation considers the influence of the amount of precipitation as the main driver of $\delta^{18}$O changes. Basic on the facts listed above, recorded in JS9 stalagmite episode of low $\delta^{18}$O and $\delta^{13}$C values can be interpreted as wet periods with more intensive vegetation.

The second part of MIS 5e (122 – 115 ka) has ca. 0.3 ‰ higher $\delta^{18}$O values and ca. 0.5 ‰ higher $\delta^{13}$C values.

Globally, this period is characterized by a systematic deterioration of climatic conditions, a global mean temperature decrease of approximately 1°C (Goelzer et al. 2016) and a sea level decrease of approximately 30 m (Grant et al., 2012). The NGRIP record shows a 6‰ decrease, which reflects the changes in thermal conditions (Rasmussen et al., 2014). At the local scale, the elevated $\delta^{13}$C value is the response to a decline in vegetation conditions. The increase in the $\delta^{18}$O value is caused by aridization. Evaporation overcomes the temperature effect,

which should be negative. A similar episode containing high $\delta^{18}$O and $\delta^{13}$C values at approximately 119 - 117 ka is observed in the records from Baradla Cave and Magurska Cave; these records are interpreted as indicating an episode of dry continental climate (Demény et al., 2017; Pawlak et al., 2020). Records from Magurska Cave, located in the Tatra Mts., and from Baradla Cave, located in Hungary (Fig. 6), are more similar to the JS9 record

during the period of 122 – 115 ka than during the older 127 – 123 ka period, which suggests that the climate of the region becomes more uniform at the end of MIS 5e.

During the 108 – 101 ka interval, the $\delta^{18}O$ and $\delta^{13}C$ values of the JS9 stalagmite were elevated, and their values were the highest in comparison to the whole recorded period, which were ca. 1.3 ‰ higher than the mean Holocene value in the DCS (Fig. 4). Elevated stable isotope values are related to elevated Fe, Mn, P, and Na contents. In the case of the JS9 stalagmite, the interpretation of the 108 – 101 ka period as a dry interval is probable and is in accordance with $\delta^{18}O$, $\delta^{13}C$, and Mg proxies. Elevated values of geochemical proxies are related to micrite microfabrics (Fig. 4 K). It appears plausible that the presence of micrite fabric (M) is indicative of bio-influenced processes, as micrite layers may be associated with shifts to more positive C isotope ratios (Kaźmierczak et al., 1996). In a sample from Nullarbor, $\delta^{13}C$ values shift from -10.5‰ to -4.0‰ in stromatolitic-like micrite (M) layers. This phenomenon was interpreted as a possible result of microbial colonization of the speleothem surface during a dry period (Frisia, 2015). According to all this information, the 108 – 101 ka interval in the JS9 stalagmite record can be interpreted as a stable dry continental climatic period.

In contrast, the global and regional situations at that time are different. After 110 ka, the decrease in the global ocean level ended. From 108 to 101 ka, the world ocean level became elevated to 20 m in comparison to the local minimum at 110 ka (Grant et al., 2012), which suggests that the global mean annual temperature increased. This trend is also in accordance with increased insolation at that time (Berger 1978). However, the global sea level was more unstable with changes reaching 10 m. Similarly, the NGRIP record has elevated $\delta^{18}O$ values with short decreasing disturbance at 105 ka, which also reflects climatic instability (Fig. 6; Rasmussen et al., 2014). The climate instability at 108 – 101 ka is also expressed by the $\delta^{18}O$ record from Magurska Cave (Fig. 6) and by records from the northern rim of the Alps (Boch et al., 2011). In contrast, the JS9 record expresses specific local stable conditions during the 108 – 101 ka period. A possible explanation is that at that time, the DCS region was constantly influenced by the continental climate, in contrast to the northern Tatra Mountains and Alps. The ca. 2‰ instant decrease and the lowest $\delta^{18}O$ value in the JS9 record at 94-95 ka are not connected with any significant change in the other measured proxies. However, this episode is expressed in the NALPS $\delta^{18}O$ records (Fig. 6) as a 1‰ instant drop and stalagmite growth cessation and is correlated with Greenland Stadial 23 (Mosley et al. 2020).

## 5. Final Conclusions

Temperature is a main factor shaping the increasing tendency in $\delta^{18}O$ values in the JS9 stalagmite in the older part of the record (143 – 130 ka). The instant change to negative values during termination II could have been caused by source and continental effects, which overcame the temperature effect at that time. The observed response is the result of both the increase in the mean annual temperature and the source/circulation effect, which overcame each other. During termination II, the records located in the western Alps and on the northern slopes of the Tatra Mountains had a positive shift in $\delta^{18}O$. This shows that mountains such as the Carpathian Belt and Alps were important climatic barriers at that time. The older MIS 5e period (127 – 123 ka) had a warmer and wetter climate. The response of proxies recorded in the JS9 stalagmite were dominated by the influence of wetter conditions. MIS 5e records from the Tatra Mts. and Baradla Cave located in Hungary (Fig. 6) are more similar to

the JS9 record during the period of 122 – 115 ka than during the older 127 – 123 ka period, which suggests that the climate of the region became more uniform at the end of MIS 5e. The climate instability at 108 – 101 ka is also expressed by the $\delta^{18}$O record from Magurska Cave (Fig. 6) and by records from the northern rim of the Alps (Boch et al., 2011). In contrast to the records from the northern rim of the Alps and Tatra Mountains (Fig. 6), the JS9 record expresses specific local stable and dry environmental conditions during the 108 – 101 ka period. A possible explanation is that at that time, the DCS region was constantly influenced by the continental climate, in contrast to the northern Tatra Mountains and Alps.

## 6. Data availability

All U-series ages used for age – depth model estimation are presented in table 1.

Estimated age-depth model, Isotopic and trace elements records data in digital form are deposited on Fighsare service

https://figshare.com/articles/dataset/Speleothem_oxygen_record_-_thermal_or_moisture_changes_proxy_A_case_study_of_multiproxy_record_from_MIS_5_MIS_6_age_speleothems_from_Dem_nov_Cave_System_/13116506

DOI: 10.6084/m9.figshare.13116506.

Other data used for comparison (fig. 6) are available  at : https://www.ncdc.noaa.gov/data-access/paleoclimatology-data/datasets and in supplementary materials of cited papers in fig. 6 caption.

## 7. Competing interests

The author declare that he has no conflict of interest

## 8. Acknowledgements.

This study was supported by a grant from the Polish Ministry of Science No-20 15/19/D/ST10/00571. U-series dating, and geochemical analyses were supported by the Plan of Institutional Financing of the Institute of Geology, The Czech Academy of Sciences (No. RVO 67985831). This research would not have been possible without a permit and help from the Tatra National Park and Slovak Caves Administration. Authors would like to thank the reviewers for their constructive comments on the manuscript.

Andersen, K., Azuma, N. and all North Greenland Ice Core Project members 2004. High-resolution record of Northern Hemisphere climate extending into the last interglacial period. Nature 431, 147–151.

Baker, A., Hartmann, A., Duan, W., Hankin, S., Comas-Bru, L., Cuthbert, M. O., Treble P. C., Banner J., Genty, D. Baldini, L.M, Bartolomé, M., Moreno, A., Pérez-Mejías, C., Werner, M., 2019. Global analysis reveals climatic controls on the oxygen isotope composition of cave drip water. Nature Communications 10, 2984.

Bar-Matthews, M., Ayalon, A., Gilmour, M., Matthews, A., Hawkesworth, C.J., 2003. Sea-land oxygen isotopic relationships from planktonic foraminifera and speleothems in the Eastern Mediterranean region and their implication for paleorainfall during interglacial intervals. Geochimica et Cosmochimica Acta 67, 3181–3199.

Boch, R., Cheng, H., Spötl, C., Edwards, R. L., Wang, X., and Häuselmann, Ph., 2011. NALPS: a precisely dated European climate record 120–60 ka, Climate of the Past 7, 1247–1259.

Bella, P., 1993. Remarks on the genesis of the Demänová Cave System. Slovenský kras 31, 43–53 (in Slovak, English abstract).

Bella, P., Gradziński, M., Hercman, H., Leszczyński, S. and Nemec, W., 2021. Sedimentary anatomy and hydrological record of relic fluvial deposits in a karst cave conduit. Sedimentology 68, 425-448.

Berger, A., 1978. Long-Term Variations of Daily Insolation and Quaternary Climatic Changes, Journal of Atmospheric Sciences 35(12), 2362-2367.

Bianchi G.G., McCave I.N., 1999. Holocene periodicity in North Atlantic climate and deep-ocean flow south of Iceland. Nature 397(6719), 515-517.

Błaszczyk, M., Hercman, H., Pawlak, J., Szczygieł, J., 2021. Paleoclimatic reconstruction in the Tatra Mountains of the western Carpathians during MIS 9–7 inferred from a multiproxy speleothem record. Quaternary Research 99, 290-304.

Chappellaz J., Brook E., Blunier T., Malaize B., 1997. CH4 and δ18O of O2 records from Antarctic and Greenland ice: A clue for strati-graphic disturbance in the bottom part of the Greenland Ice Core Project and the Greenland Ice Sheet Project 2 ice cores. Journal of Geophysical Research 102, 26547–26557.

Celle-Jeanton H, Travi Y, Blavoux B. 2001. Isotopic typology of the precipitation in the Western Mediterranean region at three different time scale. *Geophysical Research Letters* 28, 1215-1218.

Cheng, H., Edwards, R.L., Shen, C.C., Polyak, V.J., Asmerom, Y., Woodhead, J., Hellstrom, J., Wang, Y., Kong, X., Sp, C., Wang, X., Alexander, E.C., 2013. Improvements in [230]Th dating, [230]Th and [234]U half-live values, and U-Th isotopic measurements by multi-collector inductively coupled plasma mass spectrometry. Earth and Planetary Science Letters 371-372, 82-91.

Couchoud, I., Genty, D., Hoffmann, D., Drysdale, R., Blamart, D., 2009. Millennial-scale climate variability during the Last Interglacial recorded in a speleothem from south-western France, Quaternary Science Reviews 28, 3263-3274.


Comas-Bru, L., Rehfeld, K., Roesch, C., Amirnezhad-Mozhdehi, S., Harrison, S. P., Atsawawaranunt, K., Ahmad, S. M., Ait Brahim, Y., Baker, A., Bosomworth, M., Breitenbach, S. F. M., Burstyn, Y., Columbu, A., Deininger, M., Demény, A., Dixon, B., Fohlmeister, J., Hatvani, I. G., Hu, J., Kaushal, N., Kern, Z., Labuhn, I., Lechleitner, F. A., Lorrey, A., Martrat, B., Novello, V. F., Oster, J., Pérez-Mejías, C., Scholz, D., Scroxton, N.,

Sinha, N., Ward, B. M., Warken, S., Zhang, H., and the SISAL members: SISALv2 2020. A comprehensive speleothem isotope database with multiple age-depth models, Earth System Science Data 12, 2579–2606

Dansgaard, W., 1964. Stable isotopes in precipitation. Tellus 16, 436-468.

Demény, A., Kern, Z., Czuppon, G., Németh, A., Leél-Őssy, S., Siklósy, Z., Lin, K., Hu H-M, Shen, Ch-Ch., Vennemann, T.W., Haszpra, L., 2017. Stable isotope compositions of speleothems from the last interglacial – Spatial patterns of climate fluctuations in Europe, Quaternary Science Reviews 161, 68-80.

Dorale, J.A., Liu, Z., 2009. Limitations of Hendy Test criteria in judging the paleoclimatic suitability of

speleothems and the need for replication. Journal of cave and karst studies 71(1), 73–80.

Drysdale R.N., Zanchetta G., Hellstrom J.C., Fallick A.E., Zhao J., 2005. Stalagmite evidence for the onset of the Last Interglacial in southern Europe at 129 ± 1 ka. Geophysical Research Letters 32, L24708.

Droppa, A., 1957. Demänovské jaskyne. Vydavatelstvo Slovenskej Akadémie Vied.  Bratislava. (in Slovak, German summary).

Droppa, A. 1966. The correlation of some horizontal caves with river terraces. Studies in Speleology 1, 186-192.

Droppa A. 1972. Geomorfologické pomery Demänovskej doliny. Slovenský Kras 10, 9-46 (in Slovak, German summary).

Eggins, S.M., Woodhead, J.D., Kinsley, L.P.J., Mortimer, G.E., Sylvester, P., McCulloch, M.T., Hergt, J.M., Handler, M.R., 1997. A simple method for the precise determination of ≥ 40 trace elements in geological samples

by ICPMS using enriched isotope internal standardization. Chemical Geology 134, 311-326.

Elmore, A., Wright, J.D., Southon, J. 2015. Continued meltwater influence on North Atlantic Deep. Water instabilities during the early Holocene. Marine Geology 360, 17-24.

Fairchild, I.J., Baker, A., 2012. Speleothem Science: From Process to Past Environment. Willey-Blackwell ISBN:9781405196208 1-432.

Fairchild, I., Treble, P., 2009. Trace elements in speleothems as recorders of environmental change. Quaternary Science Reviews 28, 449-468.


Feurdean, A., Perşoiu, A., Tanţău, I., Stevens, T., Magyari, E.K., Onac, B.P., Marković, S., Andrič, M., Connor, S., Fărcaş, S., Gałka, M., Gaudeny, T., Hoek, W., Kolaczek, P., Kuneš, P., Lamentowicz, M., Marinova, E., Michczyńska, D.J., Perşoiu, I., Płóciennik, M., Słowiński, M., Stancikaite, M., Sumegi, P., Svensson, A, Tămaş, T., Timar, A., Tonkov, S., Toth, M., Veski, S., Willis, K.J., Zernitskaya, V., 2014. Climate variability and
associated vegetation response throughout Central and Eastern Europe (CEE) between 60 and 8 ka, Quaternary Science Reviews 106, 206 – 244.

Frisia, S. 2015. Microstratigraphic logging of calcite fabrics in speleothems as tool for palaeoclimate studies. International Journal of Speleology 44, 1-16.

Gaál, Ľ., 2016. Litológia karbonatických hornín Demänovského jaskynného systému. *Slovenský kras* 54**,** 109-129 (in Slovak, English abstract).

Gaál, Ľ., Michalík, J., 2017. Strednotriasové vápence v jaskyni Okno (Demänovská dolina, Nízke Tatry): litológia
a faciálne typy. Slovenský kras 55, 145-154 (in Slovak, English abstract).

Gascoyne, M., 1992. Paleoclimate determination from cave calcite deposits, Quaternary Science Review 11, 609-632.

Genty, D., Blamart, D., Ghaleb, B., Plagnes, V., Causse, C.h., Bakalowicz, M., Zouari, K., Chkir, N., Hellstrom, J., Wainer, K., Bourges, F., 2006. Timing and dynamics of the last deglaciation from European and North African $\delta^{13}C$ stalagmite profiles–comparison with Chinese and South Hemisphere stalagmites. Quaternary Science Review 25, 2118–2142.

Goelzer, H., Huybrechts, P., Loutre, M. F., Fichefet, T. 2016 Last Interglacial climate and sea-level evolution from a coupled ice sheet–climate model, Climate of the Past 12, 2195–2213.

Govin, A., Capron, E., Tzedakis P.C., Verheyden, S., Ghaleb, B., Hillaire-Marcel, C., St-Onge G., Stoner, J.S., Bassinot, F., Bazin, L., Blunier, T., Combourieu-Nebout, N., Ouahabi, A.E., Genty, D., Gersonde R., Jimenez-
Amat P., Landais, A., Martrat B, Masson-Delmotte V., Parrenin, F., Seidenkrantz, M.S., Veres, D., Waelbroeck, C., Zahn, R., 2015. Sequence of events from the onset to the demise of the Last Interglacial: Evaluating strengths and limitations of chronologies used in climatic archives. Quaternary Science Reviews 129, 1 – 36.

Grant, K. M., Rohling, E. J., Bar-Matthews, M., Ayalon, A., Medina-Elizalde, M., Ramsey, C. B., Satow, C.,
and Roberts, A. P., 2012. Rapid coupling between ice volume and polar temperature over the past 150 000 years. Nature 491, 744–747,

Hercman, H., Gąsiorowski, M., Pawlak, J., Błaszczyk, M., Gradziński, M., Matoušková, Š., Zawidzki, P., Bella, P., 2020. Atmospheric circulation and the differentiation of precipitation sources during the Holocene inferred
from five stalagmite records from Demänová Cave System (Central Europe). Holocene 30, 834- 846.

Hercman, H., Pawlak, J., 2012. MOD-AGE: An age-depth model construction algorithm. Quaternary Geochronology 12, 1-10.

Hercman, H., 2000. Reconstruction of palaeoclimatic changes in central Europe between 10 and 200 thousand years BP, based on analysis of growth frequency of speleothems. Studia Quaternaria 17, 35 – 70.

Hercman, H., Bella, P., Głazek, J., Gradziński, J., Lauritzen, S., Lovlie, R., 1997. Uranium series dating of speleothems from Demanova ice cave: a step to age estimation of the Demanova cave system (Nizkie Tatry MTS.,
Slovakia). Annales Societatis Geologorum Poloniae 67, 439 – 450.

Herich, P., 2017. Demänová caves. The most extensive underground karst phenomenon in Slovakia. Bulletin of the Slovak Speleological Society, Issued for the purpose of the 17th Congress of the IUS, Sydney 2017, 27-38.

Hellstrom, J., 2003. Rapid and accurate U/Th dating using parallel ion counting multicollector ICP-MS. Journal of Analytical Atomic Spectrometry 18, 1346–1351.

Hellstrom, J., 2006. U–Th dating of speleothems with high initial $^{230}$Th using stratigraphical constraint. Quaternary Geochronology1, 289–295.

Holden, N.E., 1990, Total half-lives for selected nuclides. Pure and Applied Chemistry 62, 941-958.

Holko, L., Dóša, M., Michalko, J., Šanda, M., 2012. Isotopes of oxygen-18 and deuterium in precipitation in Slovakia. Journal of Hydrology and Hydromechanics, *60*(4), 265-276.

Holzkamper, S., Mangini, A., Spotl, C., Mudelsee, M., 2004. Timing and progression of the Last Interglacial derived from a high alpine stalagmite. Geophysical Research Letter 31, L07201

Hu, C., Huang, J., Fang, N., Xie, S., Henderson, G. M., Cai, Y., 2005. Adsorbed silica in stalagmite carbonate and
its relationship to past rainfall. Geochimica et Cosmochimica Acta 69, 2285-2292.

Ionita, M., 2014. The impact of the East Atlantic/Western Russia pattern on the hydroclimatology of Europe from mid-winter to late spring. Climate 2, 296–309.

Jaffey, A.H., Flynn, K.F., Glendenin, L.E., Bentley, W.C., Essling, A.M., 1971. Precision measurement of half-lives and specific activities of $^{235}$U and $^{238}$U. Physical Review C 4, 1889-1905.

Jankovsk_a, V., Chromý, P., Ni_zniansk_a, M., 2002. _Safarka e first palaeobotanical data
on vegetation and landscape character of Upper Pleistocene in West Carpathians (North East Slovakia). Acta
Palaeobotanica 42, 29-52.

Kaźmierczak J., Coleman M.L., Gruszczyński M., Kempe S., 1996 Cyanobacterial key to the genesis of micritic and peloidal limestones in ancient seas. Acta Palaeontologica Polonica, 41, 319-338.

Kern, Z., Demény, A., Perşoiu, A., Hatvani, IG., 2019. Speleothem Records from the Eastern Part of Europe and Turkey—Discussion on Stable Oxygen and Carbon Isotopes. Quaternary. 2, 3-31.

Koltai, G., Cheng, H., Spötl, C., 2018. Paleoclimate significance of speleothems in crystalline rocks: a test case from the Late Glacial and early Holocene (Vinschgau, northern Italy). Climate of the Past 14, 369-381.


Kottek M., Grieser J., Beck Ch., Rudolf B., Rubel F., 2006. World Map of the Koppen-Geiger climate classification updated. Meteorologische Zeitschrift 15, 259–263.

Lachniet M.S., 2009. Climatic and environmental controls on speleothem oxygen-isotope values. Quaternary

Science Reviews 28, 412–432.

Lisiecki, L.E., Raymo, M.E., 2005. A Pliocene-Pleistocene stack of 57 globally distributed benthic $\delta18O$ records, Paleoceanography, 20, PA1003.

McDermott F., Atkinson T.C., Fairchild I.J., Baldini L.M., Mattey D.P. 2011. A first evaluation of the spatial gradients in $\delta^{18}O$ recorded by European Holocene speleothems. Global and Planetary Change 79, 275-287.

Meyer, M.C., Spötl, Ch., Mangini, A., 2008. The demise of the Last Interglacial recorded in isotopically dated speleothems from the Alps, Quaternary Science Reviews 27, 476-496.


Moseley, G.E., Spötl, C., Cheng, H., Boch, R., Min, A., Edwards, L.R., 2015. Termination-II interstadial/stadial climate change recorded in two stalagmites from the north European Alps, Quaternary Science Reviews 127, 229-239.

Moseley, G. E., Spötl, C., Brandstätter, S., Erhardt, T., Luetscher, M., and Edwards, R. L., 2020. NALPS19: sub-orbital-scale climate variability recorded in northern Alpine speleothems during the last glacial period, Climate of the Past 16 29–50.

Motyka, J., Gradziński, M., Bella, P., Holúbek, P., 2005. Chemistry of waters from selected caves in Slovakia – a

reconnaissance study. Environmental Geology 48, 682-692.

Nehme, C., Verheyden, S., Noble, S. R., Farrant, A. R., Sahy, D., Hellstrom, J., Delannoy, J. J., and Claeys, P. 2015. Reconstruction of MIS 5 climate in the central Levant using a stalagmite from Kanaan Cave, Lebanon, Climate of the Past 11, 1785–1799.


Pawlak, J., Błaszczyk, M., Hercman, H., Matoušková, Š., 2019. A continuous stable isotope record of last interglacial age from the Bulgarian Cave Orlova Chuka, Geochronometria *46,* 87-101.

Pawlak, J., Błaszczyk, M., Hercman, H., Matoušková, Š., 2020. Palaeoenvironmental conditions during MIS

6/MIS 5 transition recorded in speleothems from the Tatra Mountains. Boreas 50, 224 – 241.

Rasmussen, O. S., Bigler, M., Blockley, S. P., Blunier, T., Buchardt, S. L., Clausen, H. B., Cvijanovic, I, Dahl-Jensen, D., Johnsen, S. J., Fischer, H., Gkinis, V., Guillevic, M., Hoek, W. Z., Lowe, J. J., Pedro, J. B., Popp, T., Seierstad, I. K., Steffensen, J. P., Svensson A. M., Vallelonga, P., Vinther, B. M., Walker, M. J. C., Wheatley, J.

670 J. Winstrup, M., 2014. A stratigraphic framework for abrupt climatic changes during the Last Glacial period based on three synchronized Greenland ice-core records: refining and extending the INTIMATE event stratigraphy. Quaternary Science Reviews 106, 14-28.

Roberts, N., Smart, P.L., Baker A., 1998. Annual trace element variations in a holocene speleothem. Earth and
675 Planetary Science Letters 154: 237-246.

Rossi, C., Mertz-Kraus, R., Osete, M. L., 2014. Paleoclimate variability during the Blake geomagnetic excursion (MIS 5d) deduced from a speleothem record, Quaternary Science Reviews 102, 166-180.

680 Różański, K., Araguás-Araguás, L., Gonfiantini, R., 1993. Isotopic patterns in Global Precipitation. Journal of Geophysical Research Atmosphres 78, 1-36.

Rybak, O.O., Volodin, E.M., Morozova, P.A., 2018. Reconstruction of Climate of the Eemian Interglacial Using an Earth System Model. Part 1. Set–up of Numerical Experiments and Model Fields of Surface Air Temperature
685 and Precipitation Sums. Russian Meteorology and Hydrology 43, 357–365

Shin, J., Nehrbass-Ahles, C., Grilli, R., Chowdhry Beeman, J., Parrenin, F., Teste, G., Landais, A., Schmidely, L., Silva, L., Schmitt, J., Bereiter, B., Stocker, T. F., Fischer, H., Chappellaz, J. 2020. Millennial-scale atmospheric $CO_2$ variations during the Marine Isotope Stage 6 period (190–135 ka). Climate of the Past 16,
690 2203–2219.

Sotak, S., Borsanyi, P., 2004. Monitoring klimy SHMU na uzemi Nizkych Tatier. Priroda Nizkych Tatier 1, Banska Bystrica 275-282.

695 Tremaine, D., M., Froelich, P., N., 2013. Speleothem trace element signatures: A hydrologic geochemical study of modern cave drip waters and farmed calcite, Geochimica et Cosmochimica Acta 121, 522-545.

Wong, C. I., Breecker, D.O., 2015 Advancements in the use of speleothems as climate archives, Quaternary Science Reviews 127, 1-18.
Vansteenberge, S., Verheyden, S., Cheng, H., Edwards, R. L., Keppens, E., Claeys, P. 2016. Paleoclimate in continental northwestern Europe during the Eemian and early Weichselian (125–97 ka): insights from a Belgian speleothem, Climate of the Past 12, 1445–1458.


**Tables and figures**

Table 1 – U-series results.



| H*<br>[mm] | U<br>[ppm] | $^{234}U/^{238}U$<br>AR | $^{230}Th/^{234}U$<br>AR | $^{230}Th/^{232}Th$<br>AR | Age**<br>[ka] | corrected age<br>[ka] | Initial $^{234}U/^{238}U$<br>AR |
|---|---|---|---|---|---|---|---|
| 3±0.5 | 0.193±0.001 | 1.827±0.010 | 0.785±0.009 | 232±3 | 142±3 | 140±4 | 2.223±0.065 |
| 7.5±0.5 | 0.271±0.002 | 2.299±0.015 | 0.788±0.010 | 787±9 | 139±3 | 137±3 | 2.905±0.066 |
| 16±0.5 | 0.241±0.001 | 2.750±0.002 | 0.777±0.006 | 453±4 | 132±2 | 132±2 | 3.531±0.048 |
| 33±0.5 | 0.284±0.002 | 1.824±0.011 | 0.706±0.010 | 433±6 | 118±3 | 118±3 | 2.146±0.049 |
| 43±0.5 | 0.225±0.002 | 2.120±0.009 | 0.692±0.007 | 68±1 | 113±2 | 111±3 | 2.527±0.062 |
| 81±0.5 | 0.245±0.002 | 2.000±0.010 | 0.678±0.010 | 429±6 | 110±3 | 109±3 | 2.356±0.057 |
| 107±0.5 | 0.295±0.002 | 1.996±0.011 | 0.615±0.009 | 632±9 | 95±2 | 94±2 | 2.295±0.050 |
| 128±0.5 | 0.216±0.002 | 1.755±0.007 | 0.606±0.008 | 287±4 | 94±2 | 92±3 | 1.976±0.056 |
| 140±0.5 | 0.235±0.001 | 1.742±0.006 | 0.593±0.007 | 90±1 | 90±2 | 89±2 | 1.952±0.044 |
| 145±0.5 | 0.296±0.003 | 1.954±0.013 | 0.581±0.018 | 2396±73 | 87±1 | 87±1 | 2.217±0.023 |

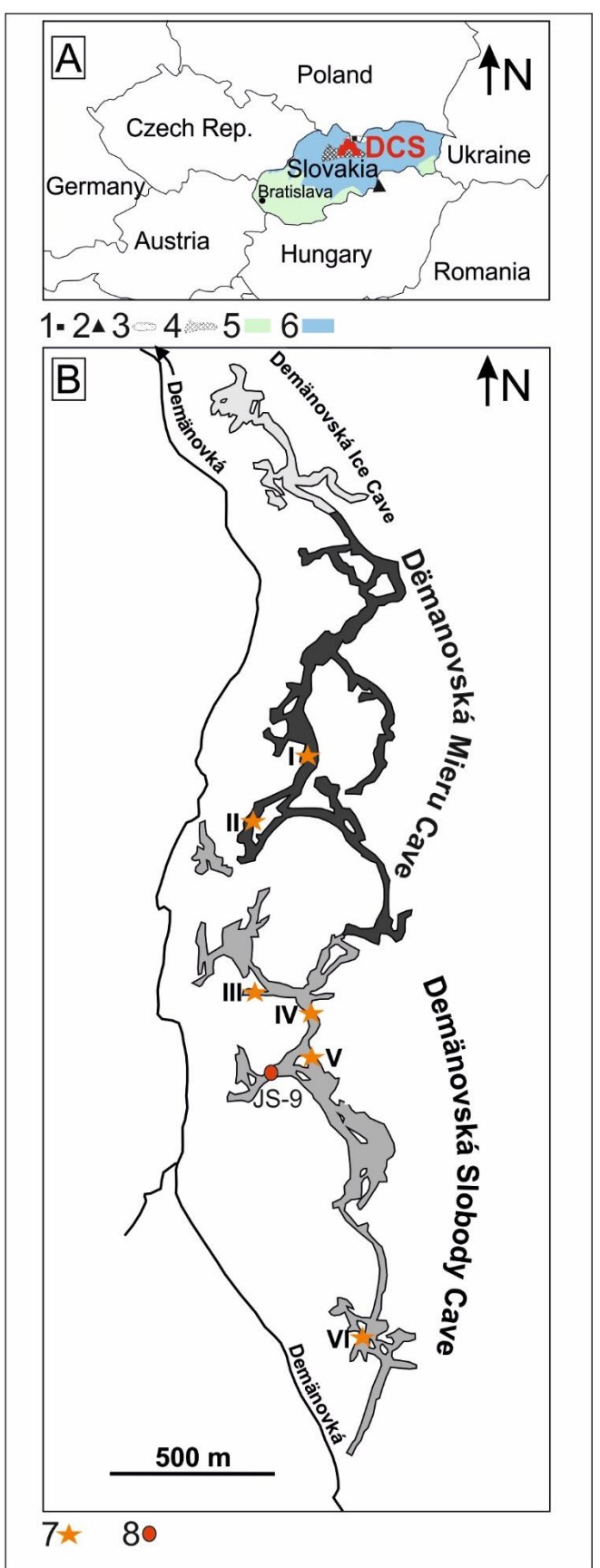

Fig 1 Local settings. A – Demänová Cave System localization, B – Map of Demänová Cave System, 1 – Magurska
Cave Site, 2 Baradla Cave Site, 3 Tatra Mts., 4 Low Tatra Mts., 5 Cfb climate zone, 6 Dfb climate zone, 7 – the
sites with cave temperature monitoring, 8 – sample collection site.

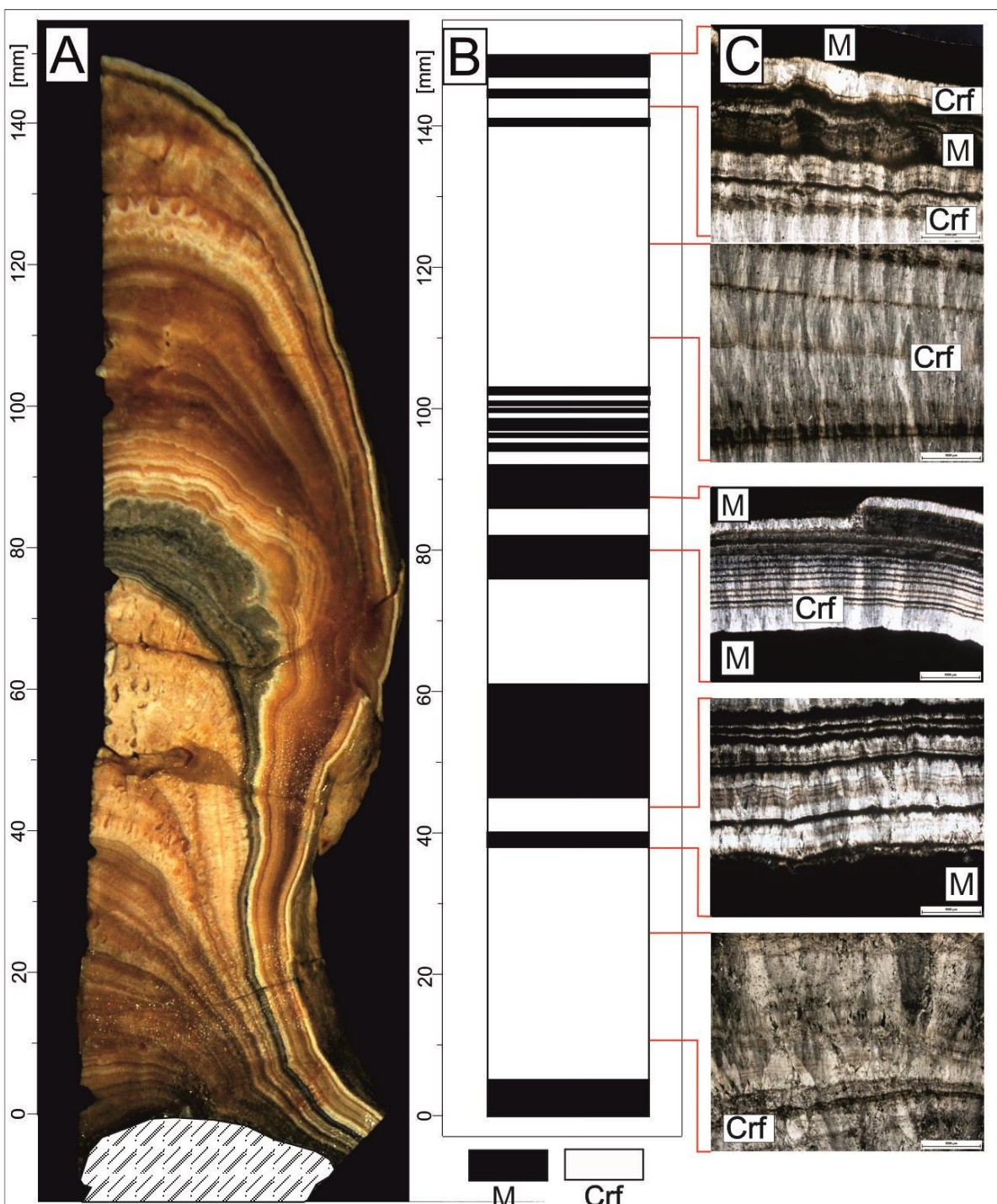

Fig 2. JS9 sample lithology. A – photo of JS9 stalagmite, B – Microfabric log in the scale of distance from the base of the stalagmite, C – Exemplary photos of microfabrics. (M) – micrite fabric, (Crf) - Columnar radiaxial fibrous microfabric.



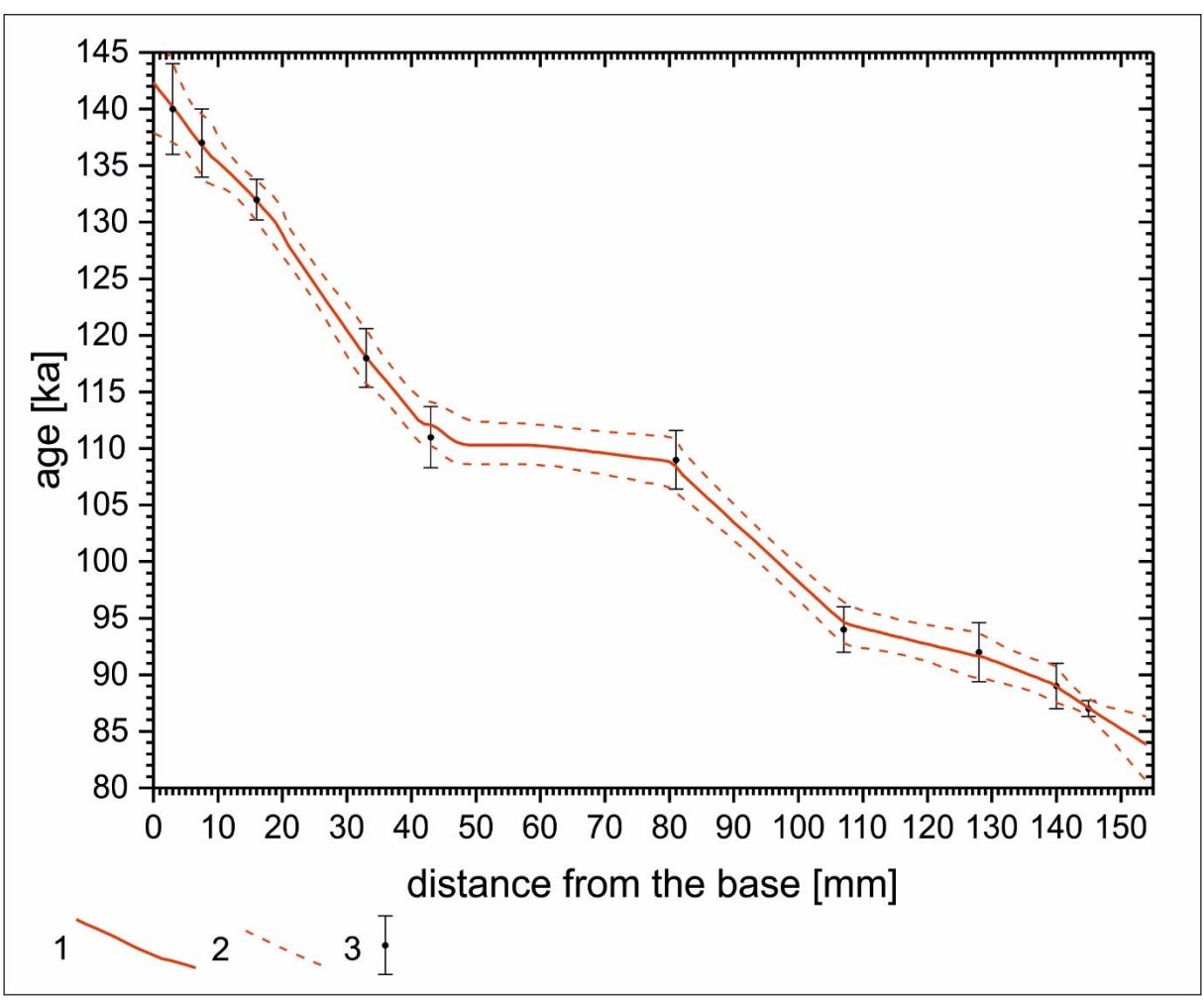

Fig 3 Age – depth model for JS9 stalagmite. 1 – age – depth model median, 2 - 2σ confidence band, 3 – U-series ages with 2σ uncertainties.




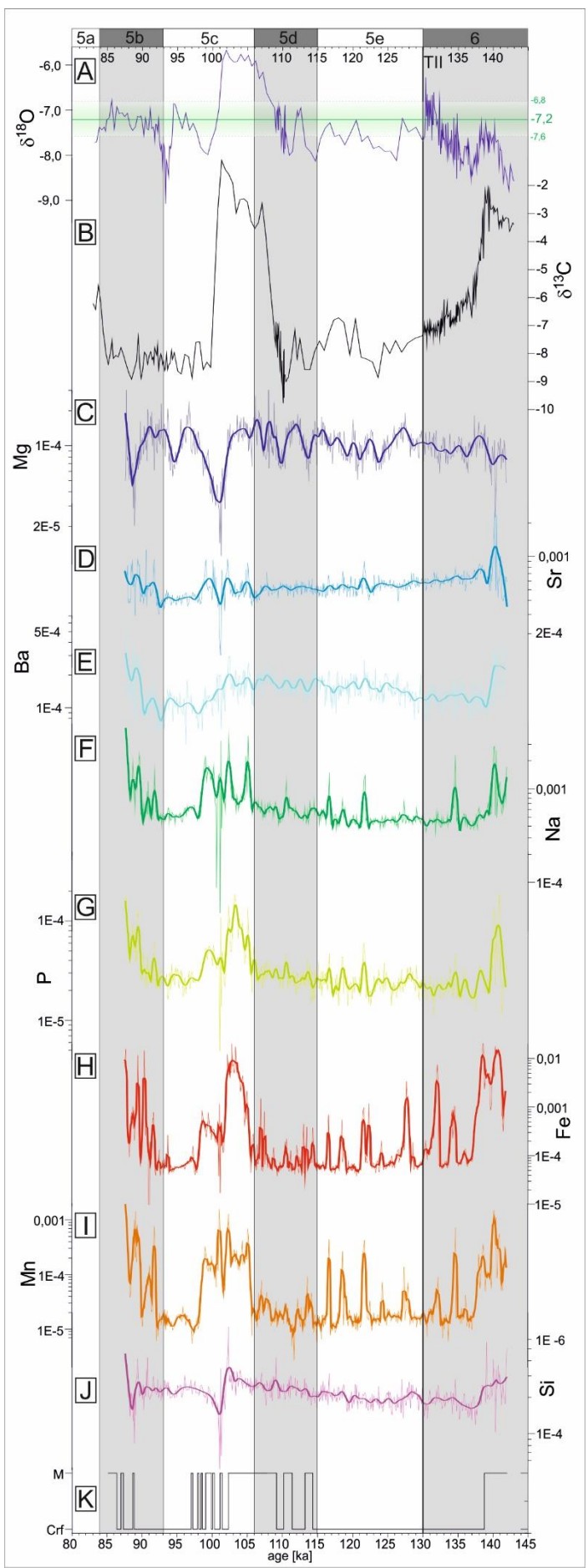

Fig. 4 Results of multi-proxy analyses of JS9 stalagmite.  A – $\delta^{18}$O composition with mean value for Holocene from DCS as a green line, B – $\delta^{13}$C composition, C – Mg content, D – Sr content, E – Ba content, F – Na content, G – P content, H – Fe content, I – Mn content, J – Si content, K – microfabrics LOG. Data availability DOI: 10.6084/m9.figshare.13116506.

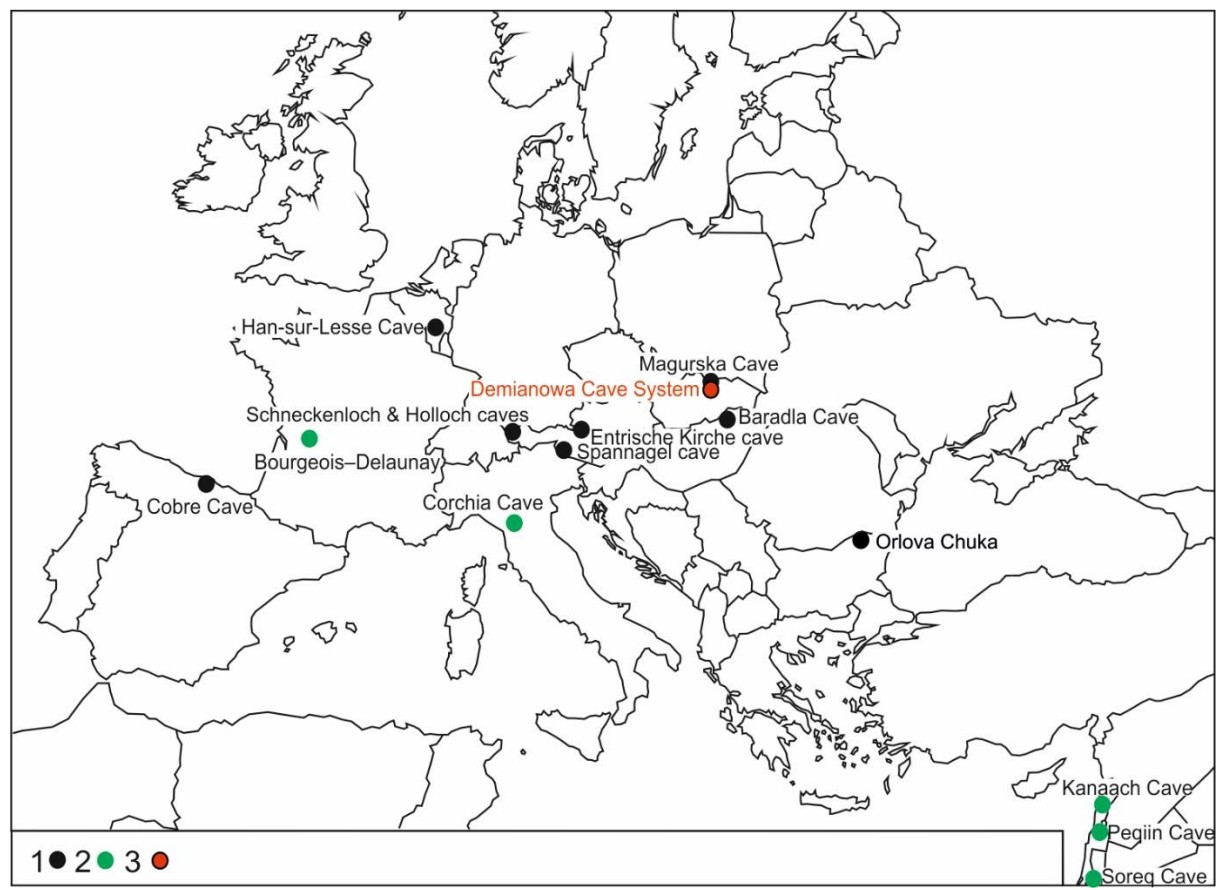


Fig. 5 Localisation of European and Middle East MIS 5/MIS 6 speleothem sites. 1 – Speleothems with temperature as a dominant factor influencing on $\delta^{18}$O value. 2 – Speleothems when the changes of the isotopic composition of rainwater and amount of precipitation are dominant factors influencing on $\delta^{18}$O value. 3 – Studied site.




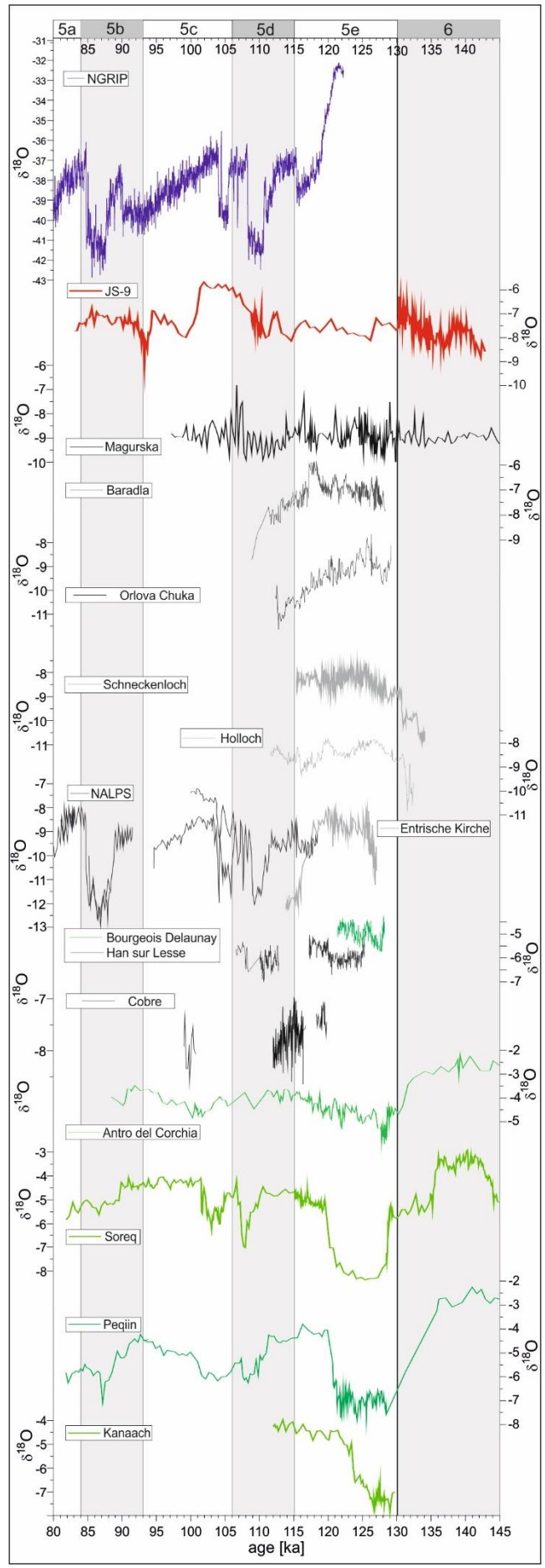

Fig. 6. The comparison of JS9 $\delta^{18}$O record with other records of MIS-5/MIS-6 age from Europe and Middle East. NGRIP (Andersen *et al.,* 2004); Magurska (Pawlak et al., 2020); Baradla (Demény et al., 2017); Orlova Tchuka (Pawlak et al., 2019); Schneckenloch (Mosley *et al.,* 2015); Holloch (Moseley *et al.,* 2015); Entrische Kirche (Meyer *et al.,* 2008); Bourgeois-Delaunay (Couchoud et al. 2009); Cobre (Rossi et al. 2014); Han-sur-Lesse (Vansteenberge et al., 2016); Antro del Corchia (Drysdale *et al.,* 2005); Soreq (Bar-Matthews *et al.,* 2003); Peqiin (Bar-Matthews *et al.,* 2003); Kanaan (Nehme *et al.,* 2015); black and gray colors charts - speleothems with temperature as a dominant factor influencing on $\delta^{18}$O value; Green colors charts – speleothems, where the changes of the isotopic composition of rainwater and amount of precipitation are dominant factors influencing on $\delta^{18}$O value; Red color charts - studied site.