# Peer review of "The speleothem oxygen record – a proxy for thermal or moisture changes? A case study of multiproxy records from MIS 5/MIS 6 speleothems from the Demänová Cave System."

_Climate of the Past, 2020_

## Short Comment (SC1) · 27 Oct 2020

This is a short comment to point out that the folding placed a younger, repeated sequence at the base of the GRIP borehole. The author should instead rely on the North GRIP and NEEM ice cores in this temporal window. To my knowledge, Stage-6 ice has not been discovered on Greenland.

---

## Referee Comment (RC1) · Anonymous Referee #1 · 18 Nov 2020

Jacek Pawlak presents a new speleothem record from Slovakia that covers Termination 2 and the last Interglacial. The record is composed of multiple geochemical proxies (stable O and C isotopes, trace elements, and carbonate microfabrics) and supported by a U-Th chronology.

The record is new and of good quality, but I think the discussion of the results and placement in the regional and temporal context needs substantial more work before being accepted in CP.

I found it quite difficult to follow the discussion in several instances. Part of this might be related to wording and language, I made some suggestions for improvement but more could be done to clarify the text. In addition, I think the figures 4 and 6 can be improved as they are quite hard to read at the moment. I made some suggestions in the specific comments.

I would suggest reorganising the discussion along three main sub-headings: 1) Drivers of d18O in Slovakia and comparison with other European d18O records. Here, I found it difficult to understand what is meant by "thermal" control, as d18O can be influenced by temperature in different ways, which the author describes, but then does not further elaborate on. More clarity in the language (what is meant by "thermal" or "temperature" effect in d18O at the different locations described?) and a more in-depth discussion of the likely controls on d18O at the DCS cave is needed. For example, is there monitoring data from the cave that can back up some of the interpretation of d18O? Nearby GNIP stations that can be used to test the modern relationship between temperature, moisture source, and d18O in precipitation? So far, this part of the discussion is tenuous and seems based mostly on speculation.

2) Interpretation of the other proxies. Again, I would welcome some more detailed reasoning on why the proxies are interpreted in the way they are. For d13C, it would be useful to know more about the soil cover and vegetation assemblage above the cave. Are there palynological/palaeoecological studies from the region that could shed light on expected changes in biosphere responses over glacial-interglacial timescales? The discussion of the trace elements similarly lacks depth. It is not clear to me whether the author is implying a control of prior calcite precipitation (PCP) control on Mg, Sr, and Ba, or whether the dominant control is thought to be the dolomite dissolution. In any case, this should be elaborated: what is the basis for the claim that dolomite dissolution is dominant? Are there measurements of the host rock composition? Overall, I think there is currently some overinterpretation of small wiggles in the trace element records (Mg, Sr, Ba), in particular, I don't see much variability in the older part of the Mg record.

[Figure]

I would also caution against interpreting the trace element record at the base of the stalagmite, as this is often a region where effects that have nothing to do with climate play a role.

3) The temporal evolution of the proxies in JS9. In my opinion, the records are overinterpreted at this point. Many of the smaller wiggles are probably not resolvable given the chronological uncertainty in the record. I would suggest the author focus and expand the discussion on the larger and interesting features in the record, e.g., the large peak in d13C, P, Fe, and Mn around 100 ka BP, as well as why TII is only weakly expressed in the record. It is possible that with some restructuring of the text this is already possible, but as mentioned above I found it quite difficult to follow at the moment. I think it would be better to use a more recent ice core record (NGRIP for example) to compare the speleothem records to.

Specific comments: - line 27 and rest of text: use "precipitation" instead of "rain precipitation"

- line 44 and following: I would not include Middle Eastern records in the discussion about European records. Rather discuss the European records and then add a sentence showing the similarities with the Middle East and linking that to the prevailing circulation patterns?

- line 52: I don't understand this sentence.

- line 55: Since it's a single author paper, change the "we" to first person.

- line 67: I think it should be "genesis of DCS" instead of "genese"

- line 78 and in other parts of the text: "peak" instead of "pick"

- line 95: Why were samples for dating drilled to be as thick as possible? This seems counterintuitive, as one would typically try and minimise the amount of sample to avoid integrating too much time within a single sample.

- line 117: "To minimalize the difference in resolution between the lower and upper part of the studied record caused by the sedimentation rate, which is slower for the lower part. The lower part of a stalagmite from 0 to 40 mm was additionally sampled with a resolution of one sample/0.3 mm." I think these two sentences should be joined, as the first one does not make sense on its own.

- line 118: Use "growth rate" instead of "sedimentation rate" for speleothems

- line 212 and following: instead of "short time signal" it would be clearer to refer to "short term variability".

- line 233: I think it's interesting that the TII is only visible as such a muted response in d18O, compared to the overall variability in the record. I would be interested in knowing more about why that is.

- line 255: The growth rate appears to be much lower in the interval 127-122, which stands at odds with the interpretation of the isotopes (wetter climate and well-developed soil). Any thoughts on why that might be? Also, I don't agree with the following sentence on Mg, as I don't think there is a significant trend there.

- Figure 1: the map of Slovakia could be improved by showing the topography (Tatra mountains) and location of Magurska Cave, plus the boundaries of climate zones or dominant air masses.

- Figures 4 and 6: I find these two figures very hard to read, as there is a lot of information. Given that they are so crucial to the discussion, I have a few suggestions to improve them: 1. I would add a second age scale at the top of the figures, to make it easier to read 2. Clearly label the TII, maybe as a bar that covers the figure 3. I wonder if it is necessary to show all the records in figure 6, I think focusing on some key records from each group (maybe the longest ones?) would make it easier to read. But this might be my personal preference. I like the colour scheme linking back to figure 5, please add the explanation of the different groups to the caption.

---

## Referee Comment (RC2) · Anonymous Referee #2 · 25 Nov 2020

The manuscript submitted to CP by Jacek Pawlak discusses an interesting multi-proxy speleothem record from Slovakia that spans MIS6/MIS6.

The new paleoclimate data are very interesting, however, the manuscript is poorly written. The manuscript needs some deep re-structuring/re-writing. The English language style might benefit from a language editor.

The manuscript has a high potential for CP after being revised thoroughly.

My comments are list below:

[Figure]

- Abstract: 1- something is missing before you start presenting the JS9 stalagmite. Please present the 'problematic', the questions that you're trying to answer, before talking about JS9. 2- please clarify what you mean by "transitional and continental climate" 3- why do you have to mention "in opposition to the records from the Alps and the northern Tatra mountains" in the abstract? if it is so important, then please explain what you concluded about this difference with the Alps and Tatra..

- Introduction: 4- replace 'most suitable" with "most commonly used" 5- line 30: references are missing after "nordic seas" and "Atlantic ocean". 6- line 40: please explain what do you mean by stating that the speleothem 18O can be influenced by PCP? and add references to the new statement as well. As far as I know, PCP influences mostly 13C and not 18O, but I might be wrong. Please check... 7- between line 55 and 60: something is missing before you write "we present ca. 60...". Please state why a new speleothem record is needed before you present it. 8- before line 85: "several generations of speleothems" doesn't seem like a correct expression here.

- Methods: 9- write "in terms of" instead of "in a term of" 10- line 95: typo "oof" 11- the steps described between line 95 and 100 require some re-writing 12- line 105: "calculated by taking into account" instead of "with taking in the account" 13- before line 110: "taken into account" instead of "take" 14- line 110: "modified" instead of "changed" 15- after line 115: "minimize" instead of "minimalize"

-Results: 16- line 145: "described by Frisia (2015)" instead of "in the work of" 17- same as before (line 150). 18- "the" used procedure 19- line 160: Helltrom's procedure. a reference is needed here 20- "relatively" slow instead of relative 21- between lines 170 and 175, replace "since" with "from" whenever you refer to time periods. 22- line 185: write "similar" instead of "like each other" 23- replace "at that time too" with " during the same period". 24- general comment: refer to a figure whenever you need to mention information related to specific time periods.

- Discussion: 25- general comment: the main results of this paper are not highlighted

in a sufficient way. The main conclusions and findings need to be well presented. 26-replace "exemplary" with "for example" throughout the manuscript 27- replace "in opposition" with "contrariwise or on the opposite" throughout the manuscript 28- replace "the JS9 stalagmite with "the stalagmite JS9" throughout the manuscript 29- replace "dryer" with "drier" throughout the manuscript

- Conclusion: 30- I would rather write the conclusion in the form of a paragraph instead of bullet points.

---

## Author Comment (AC1) · 19 Jan 2021

Jacek Pawlak presents a new speleothem record from Slovakia that covers Termination 2 and the last Interglacial. The record is composed of multiple geochemical proxies (stable O and C isotopes, trace elements, and carbonate microfabrics) and supported by a U-Th chronology.

[Figure]

The record is new and of good quality, but I think the discussion of the results and placement in the regional and temporal context needs substantial more work before being accepted in CP.

I found it quite difficult to follow the discussion in several instances. Part of this might be related to wording and language. I made some suggestions for improvement but more could be done to clarify the text. In addition, I think the figures 4 and 6 can be improved as they are quite hard to read, at the moment. I made some suggestions in the specific comments:

I would suggest reorganizing the discussion along three main sub-headings:

1) Drivers of $\delta$18O in Slovakia and comparison with other European $\delta$18O records. 2) Interpretation of the other proxies. 3) The temporal evolution of the proxies in JS9.

I agree that such rearranging of the discussion may improve clearness of the manuscript.

1) Drivers of $\delta$18O in Slovakia and comparison with other European $\delta$18O records.

Here, I found it difficult to understand what is meant by "thermal" control, as $\delta$18O can be influenced by temperature in different ways, which the author describes, but then does not further elaborate on. More clarity in the language (what is meant by "thermal" or "temperature" effect in $\delta$18O at the different locations described?)

and a more in-depth discussion of the likely controls on $\delta$18O at the DCS cave is needed. For example, is there monitoring data from the cave that can back up some of the interpretation of $\delta$18O? Nearby GNIP stations that can be used to test the modern relationship between temperature, moisture source, and $\delta$18O in precipitation? So far, this part of the discussion is tenuous and seems based mostly on speculation.

I agree that this dependence is complicated. I believe that it is more the local thermal signal (changes in mean annual temperature) which influences the $\delta$18O of precipitation and has impact on isotopic fractionation during calcite crystallization in the cave.

However, in longer time scale the isotopic composition of ocean source which depend on mean global temperature and global volume of glaciers becomes important (source effect). Additionally the circulation changes which causes the changes in proportion of vapors from different sources plays important role here.

There are several stations in Slovakia, where the $\delta$18O of precipitation was measured continuously (Holko et al 2012). The detailed study in the region shows, that there is a strong dependence between the $\delta$18O and temperature. This dependence is observed in the long time monitoring for single sites (Holko et al 2012). Exemplary, for the closest station next to the research area in Liptovski Mikulas the R2 of correlation between the $\delta$18O of atmospheric precipitation and temperature is 0.639 (Holko et al 2012). Additionally, there is a stronger dependence between mean annual temperature of the site, and mean $\delta$18O value in Slovakia with R2= 0.728 (Holko et al 2012). The dependence between mean annual precipitation and mean $\delta$18O value is less visible (R2=0.483) (Holko et al 2012).

There is other methodological work which includes data from many cave sites located on several continents the conclusion of this work is that sites with mean annual temperature lower than 15C and the aridity index higher that 0.65 has a potential for growing speleothems which reflects the mixed signal of past temperature and past precipitation ( Baker et al 2018).

All this information will be included in revised version of discussion.

2) Interpretation of the other proxies. Again, I would welcome some more detailed reasoning on why the proxies are interpreted in the way they are. For $\delta$13C, it would be useful to know more about the soil cover and vegetation assemblage above the cave. Are there palynological/palaeoecological studies from the region that could shed light on expected changes in biosphere responses over glacial-interglacial timescales? The discussion of the trace elements similarly lacks depth. It is not clear to me whether the author is implying a control of prior calcite precipitation (PCP) control on Mg, Sr, and

Ba, or whether the dominant control is thought to be the dolomite dissolution. In any case, this should be elaborated: what is the basis for the claim that dolomite dissolution is dominant? Are there measurements of the host rock composition? Overall, I think there is currently some overinterpretation of small wiggles in the trace element records (Mg, Sr, Ba), in particular. I don't see much variability in the older part of the Mg record. I would also caution against interpreting the trace element record at the base of the stalagmite, as this is often a region where effects that have nothing to do with climate play a role.

Presently the vegetation cover over the Demianovska Cave System is dominated by the mixed forest of mountain type and grasslands connected with mountain slopes activity. Therefore, presently there are two types of soils, over the cave, associated with forest and the other type associated with slopes activity (Hercman et al 2020).

There are few palynological sites nearby. The important one is located in a closed proximity is Safarka (Jankovska, 2002). The pollen record from Safarka covers ca. last 60ka and recording changes of plants cover during the last glacial. The record show a predominance of needle-leaved and cold temperate tree vegetation during warmer episodes of MIS 3. The Last Glacial Maximum part of these pollen record suggests that some woody cover was maintained in the region during the maximum northern hemisphere ice extent (between 26.5 and 19 ka cal BP). The beginning of Holocene is marked by introduction of temperate forest (Feurdean et al 2014). The changes in vegetation at the end of last glacial and the begging of Holocene are well expressed in $\delta$13C records from Demianova Cave System (Hercman et al 2020).

The Demianova Cave is developed mostly in Gutenstein limestones (Early Anisian) and Ramsau dolomite (Ladinian) therefore both the dissolution of limestones and dolomites can happened. During periods of longer water residence time the contribution of Magnesium from Dolomite source is increased. This effect results in higher Mg/Ca values and lover Sr/Ca values (Tremaine and Froelich 2013). The PCP also can occur during dry episodes of longer water residence times. However, the PCP results in increase of

all X/Ca rations (Tremaine and Froelich 2013) which is rather not observed in a studied record.

3) The temporal evolution of the proxies in JS9. In my opinion, the records are overinterpreted at this point. Many of the smaller wiggles are probably not resolvable given the chronological uncertainty in the record. I would suggest the author focus and expand the discussion on the larger and interesting features in the record, e.g., the large peak in $\delta$13C, P, Fe, and Mn around 100 ka BP, as well as why TII is only weakly expressed in the record. It is possible that with some restructuring of the text this is already possible, but as mentioned above I found it quite difficult to follow at the moment. I think it would be better to use a more recent ice core record (NGRIP for example) to compare the speleothem records to.

I agree with your commend about overinterpretation of some part of that record in revised version of the Manuscript the discussion will be oriented more on the most important fact like the large event around 100 ka visible in this record and the expression of TII.

In previous version I decided to use GRIP due to the fact, that it covers whole studied period. However, after I got several commends and suggestions about that I agree that the NGRIP is better solution even if it covers only part of the studied record.

Specific comments: - line 27 and rest of text: use "precipitation" instead of "rain precipitation"

It will be corrected in revised manuscript.

- line 44 and following: I would not include Middle Eastern records in the discussion about European records. Rather discuss the European records and then add a sentence. showing the similarities with the Middle East and linking that to the prevailing circulation patterns?

It has been changed.

[Figure]

In contrast to the most of European records, the records from Middle-East seems to be influenced by more factors, like the amount of precipitation, temperature, and also changes in the main source of vapor for rain precipitation (source effect; Bar-Matthews et al., 2003). It can be linked to changes in the prevailing circulation patterns, the impact of evaporation on Mediterranean Sea surface $\delta$18O and also lower amplitude of long time mean annual temperature changes during Last Interglacial at lower latitudes (Rybak et al. 2018).

- line 52: I don't understand this sentence.

It has been changed to

Presently, Slovakia is influenced by two main types of climates (Kottek et. al. 2006), the boreal fully humid with warm summers climate (Dfb) on the East and warm temperate fully humid climate (Cfb) on the West.

- line 55: Since it's a single author paper, change the "we" to first person.

It will be corrected in revised manuscript.

- line 67: I think it should be "genesis of DCS" instead of "genese"

It will be corrected in revised manuscript.

- line 78 and in other parts of the text: "peak" instead of "pick"

It will be corrected in revised manuscript.

- line 95: Why were samples for dating drilled to be as thick as possible? This seems counterintuitive, as one would typically try and minimize the amount of sample to avoid integrating too much time within a single sample.

It is a mistake it should be thin here.

- line 117: "To minimalize the difference in resolution between the lower and upper part of the studied record caused by the sedimentation rate, which is slower for the lower

part. The lower part of a stalagmite from 0 to 40 mm was additionally sampled with a resolution of one sample/0.3 mm." I think these two sentences should be joined, as the first one does not make sense on its own.

The lower part of a stalagmite ( 0 to 40 mm) was additionally sampled with a resolution of one sample/0.3 mm, to minimalize the difference in resolution between the lower and upper part of the studied record caused by, slower for the lower part sedimentation rate.

- line 118: Use "growth rate" instead of "sedimentation rate" for speleothems

It will be corrected in revised manuscript.

- line 212 and following: instead of "short time signal" it would be clearer to refer to "short term variability".

Thank you, a lot, for all grammar commends they will be corrected. Additionally, the text will be sent for final language and grammar corrections. After all substantive changes and corrections.

- line 233: I think it's interesting that the TII is only visible as such a muted response in d18O, compared to the overall variability in the record. I would be interested in knowing more about why that is.

I agree. Additional question is why the final negative response of $\delta$18O at TII is opposite to response observed in other Central European sites exemplary in northern part of Tatra Montains (Magurska Cave) and in Alpine records (Holloch and Schneckenloch). The response in those Central European sites is clearly connected, with two processes first with increase of mean annual temperature and second with change from winter dominated precipitation to summer dominated precipitation (Moseley et al. 2015; Meyer et al., 2008; Holzkamper et al., 2004). In case of studied site, the observed final negative response must be caused by local or regional effect which was stronger that thermal effect at that time. The possible effect which my cause the lower

value of $\delta$18O is circulation effect and change from sources of precipitation like Adriatic Sea or Black Sea to Atlantic Source and vapor recycled over the European continent. The instant change to more negative values may be caused by source and continental effect, which overcame the temperature effect at that time. Therefore, the muted response is a result of both the effect of increase the mean annual temperature and source/circulation effect which overcomes each other.

- line 255: The growth rate appears to be much lower in the interval 127-122, which stands at odds with the interpretation of the isotopes (wetter climate and well-developed soil). Any thoughts on why that might be? Also, I don't agree with the following sentence on Mg, as I don't think there is a significant trend there.

The interval of low growth rate is much longer it starts at ca. 133 ka and ends at ca. 112. In my opinion it may be caused by local effects. Our research of five Holocene stalagmites from Demianova Cave System shows that changes in growth rate are different for all studied stalagmites and they are not connected directly with climate changes (Hercman et al., 2020). Therefore, they must be caused by local effect in the cave.

I agree with your opinion that trace elements like Mg must be treated with caution. It is true that the Mg record is dominated rather by pseudo cyclic changes without the significant main trend. However, there are two short intervals where the Mg content is significantly lower at ca. 100 and 88 ka, which is interesting. Additionally, in my opinion there are several intervals of lower Mg values which repeats $\delta$13C signal at 88, 100, 110, 113 and 124 ka. In my opinion, it is an argument that those intervals may relate to increased amount of precipitation.

- Figure 1: the map of Slovakia could be improved by showing the (Tatra Mountains, Low Tatra moununtains, location of Magurska Cave and the boundaries of climate zones or dominant air masses.

The boundaries of climate zones, locations of Tatra Mts and Low Tatra Mts and the caves in the nearest proximity (Magurska and Baradla ) have been added to fig 1A.

- Figures 4 and 6: I find these two figures very hard to read, as there is a lot of information. Given that they are so crucial to the discussion, I have a few suggestions to improve them: 1. I would add a second age scale at the top of the figures, to make it easier to read 2. Clearly label the TII, maybe as a bar that covers the figure 3. I wonder if it is necessary to show all the records in figure 6, I think focusing on some key records from each group (maybe the longest ones?) would make it easier to read. But this might be my personal preference.

According to your suggestion I added the additional age scale on the top of both figures and solid line for TII and dotted lines between MIS 5e/5d/5c/5b boundaries.

I like the color scheme linking back to figure 5, please add the explanation of the different groups to the caption.

Thank you. The explanation of the color scheme has been added to the caption bellow fig 6.

Fig. 6. The comparison of JS9 $\delta$18O record with other records of MIS-5/MIS-6 age from Europe and Middle East. GRIP (Chappellaz et al.,1997); Magurska (Pawlak et al., 2020 – submitted); Baradla (Demény et al., 2017); Orlova Tchuka (Pawlak et al., 2019); Schneckenloch (Mosley et al., 2015); Holloch (Moseley et al., 2015); Entrische Kirche (Meyer et al., 2008); Bourgeois-Delaunay (Couchoud et al. 2009); Cobre (Rossi et al. 2014); Han-sur-Lesse (Vansteenberge et al., 2016); Antro del Corchia (Drysdale et al., 2005); Soreq (Bar-Matthews et al., 2003); Peqiin (Bar-Matthews et al., 2003); Kanaan (Nehme et al., 2015); black and gray colors charts - speleothems with temperature as a dominant factor influencing on $\delta$18O value; Green colors charts – speleothems, where the changes of the isotopic composition of rainwater and amount of precipitation are dominant factors influencing on $\delta$18O value; Red color charts - studied site.

Baker, A., Hartmann, A., Duan, W. et al. 2019. Global analysis reveals climatic controls on the oxygen isotope composition of cave drip water. Nature Communications 10, 2984. https://doi.org/10.1038/s41467-019-11027-w

A. Feurdean, A. Perşoiu, I. TanÅčău, T. Stevens, E.K. Magyari, B.P. Onac, S. Marković, M. Andrič, S. Connor, S. Fărcaş, M. Gałka, T. Gaudeny, W. Hoek, P. Kolaczek, P. Kuneš, M. Lamentowicz, E. Marinova, D.J. Michczyńska, I. Perşoiu, M. Płóciennik, M. Słowiński, M. Stancikaite, P. Sumegi, A. Svensson, T. Tămaş, A. Timar, S. Tonkov, M. Toth, S. Veski, K.J. Willis, V. Zernitskaya, 2014. Climate variability and associated vegetation response throughout Central and Eastern Europe (CEE) between 60 and 8 ka, Quaternary Science Reviews;106, 206-224,

Hercman H, GÄĚsiorowski M, Pawlak J, et al. 2020. Atmospheric circulation and the differentiation of precipitation sources during the Holocene inferred from five stalagmite records from Demänová Cave System (Central Europe). The Holocene. 30(6):834-846. doi:10.1177/0959683620902224

Holko, L., Dóša, M., Michalko, J.,  Šanda, M. 2012. Isotopes of oxygen-18 and deuterium in precipitation in Slovakia / Izotopy kyslíka-18 A deutéria v zrážkach na Slovensku, Journal of Hydrology and Hydromechanics, 60(4), 265-276. doi: https://doi.org/10.2478/v10098-012-0023-2

Jankovska, V., Chromá, P., Niznianska, M., 2002. Safarka - first palaeobotanical data on vegetation and landscape character of Upper Pleistocene in West Carpathians (North East Slovakia). Acta Palaeobotanica 42, 29-52.

Tremaine D M, Froelich P N, 2013. Speleothem trace element signatures: A hydrologic geochemical study of modern cave drip waters and farmed calcite, Geochimica et Cosmochimica Acta 121; 522-545.

Please also note the supplement to this comment:
https://cp.copernicus.org/preprints/cp-2020-125/cp-2020-125-AC1-supplement.pdf

---

## Author Comment (AC2) · 19 Jan 2021

The manuscript submitted to CP by Jacek Pawlak discusses an interesting multi-proxy speleothem record from Slovakia that spans MIS6/MIS6.

The new paleoclimate data are very interesting, however, the manuscript is poorly written. The manuscript needs some deep re-structuring/re-writing. The English language style might benefit from a language editor.

The manuscript has a high potential for CP after being revised thoroughly.

My comments are list below: - Abstract:

1- something is missing before you start presenting the JS9 stalagmite. Please present the 'problematic', the questions that you're trying to answer, before talking about JS9.

This part has been rearranged.

Presently the region of central Europe is in transitional climate zone under influence of both oceanic and continental climate and continental climate. However, in the past, the region could be under stronger influence of the continental climate during cold glacial episodes or under stronger influence of oceanic climate during wetter interglacials. The long time speleothem records can adds new helpful data about past climate changes in the region. The multiproxy record of the JS9 stalagmite, collected in Demänová Cave System (Slovakia), represents ca. 60 ka period (143 – 83 ka).

2- please clarify what you mean by "transitional and continental climate"

The whole sentence has been rearranged

Presently the region of central Europe is in transitional climate zone under influence of both oceanic and continental climate and continental climate. However, in the past, the region could be under stronger influence of the continental climate during cold glacial episodes or under stronger influence of oceanic climate during wetter interglacials.

3- why do you have to mention "in opposition to the records from the Alps and the northern Tatra mountains" in the abstract? if it is so important, then please explain what you concluded about this difference with the Alps and Tatra..

In opposition to the records from the Alps and the northern Tatra mountains, the $\delta$18O record of JS9 has instant decrease episodes during Termination II. It shows that

Carpathian Belt was important climatic barrier at that time.

- Introduction:

4- replace 'most suitable" with "most commonly used"

It will be corrected in revised manuscript.

5- line 30: references are missing after "nordic seas" and "Atlantic ocean".

It has been updated

The other potential sources are the Mediterranean Sea, the Black Sea, and Nordic Seas (Ionita, 2014)

6- line 40: please explain what do you mean by stating that the speleothem 18O can be influenced by PCP? and add references to the new statement as well. As far as I know, PCP influences mostly 13C and not 18O, but I might be wrong. Please check...

You are right, It affect the $\delta$13C. It is a mistake and will be corrected in revised manuscript

7- between line 55 and 60: something is missing before you write "we present ca. 60...". Please state why a new speleothem record is needed before you present it.

It has been rearranged

However, in the past, the local climate could be more continental during colder and dryer glacial periods and more transitional at warmer interglacial periods. The new long time speleothem records can adds new helpful data about past climate changes in this region. We present ca. 60 ka long multiproxy record ($\delta$18O, $\delta$13C, Mg, Sr, Ba, Na, P, Fe, Mn, Si) of MIS-5/MIS-6 age stalagmite collected in the Demänová Cave System which is located in Slovakia.

8- before line 85: "several generations of speleothems" doesn't seem like a correct expression here.

It has been changed to:

several stages of speleothems crystallization

- Methods:

9- write "in terms of" instead of "in a term of"

It will be corrected in revised manuscript.

10- line 95: typo "oof"

It will be corrected in revised manuscript.

11-the steps described between line 95 and 100 require some re-writing

It has been rewritten to

Due to control the efficiency of chemical procedure,aAt its the beginning the spike (233U, 236U and 229Th) was added into the samples. At first step of chemical procedure, the samples were heated up for the decomposition of potential organic matter. After that the samples were dissolved in nitric acid. Finally the uranium, and the thorium were separated from the solution by chromatographic method using TRU Resin (Hellstrom, 2003).

12- line 105: "calculated by taking into account" instead of "with taking in the account"

It will be corrected in revised manuscript.

13- before line 110: "taken into account" instead of "take"

It will be corrected in revised manuscript.

14- line 110: "modified" instead of "changed"

It will be corrected in revised manuscript

15- after line 115: "minimize" instead of "minimalize"

It will be corrected in revised manuscript

-Results:

16- line 145: "described by Frisia (2015)" instead of "in the work of"

It will be corrected in revised manuscript

17- same as before (line 150).

It will be corrected in revised manuscript

18- "the" used procedure

It will be corrected in revised manuscript

19- line 160: Helltrom's procedure. A reference is needed here

The used procedure considers the possibility of contamination not only by 230Th like in original Hellstrom's procedure (Hellstrom, 2006) but also by 234U and 238U (Błaszczyk et al., 2020).

20- "relatively" slow instead of relative

It will be corrected in revised manuscript

21- between lines 170 and 175, replace "since" with "from" whenever you refer to time periods.

It will be corrected in revised manuscript.

22- line 185: write "similar" instead of "like each other"

It will be corrected in revised manuscript

23- replace "at that time too" with " during the same period".

It will be corrected in revised manuscript

24- general comment: refer to a figure whenever you need to mention information related to specific time periods.

It will be corrected in revised manuscript

- Discussion: 25- general comment: the main results of this paper are not highlighted in a sufficient way. The main conclusions and findings need to be well presented.

In the revised version of manuscript discussion is focused more on the topics like why the response on $\delta18O$ to TII is different than in other records from Central Europe and possible explanations. The effect which is visible on all proxies around 100 ka and possible explanations.

26-replace "exemplary" with "for example" throughout the manuscript

It will be corrected in revised manuscript

27- replace "in opposition "with "contrariwise or on the opposite" throughout the manuscript

It will be corrected in revised manuscript

28- replace "the JS9 stalagmite" with "the stalagmite JS9" throughout the manuscript

It will be corrected in revised manuscript

29- replace "dryer" with "drier" throughout the manuscript

It will be corrected in revised manuscript

- Conclusion: 30- I would rather write the conclusion in the form of a paragraph instead of bullet points.

Ionita, M., 2014 The impact of the East Atlantic/Western Russia pattern on the hydroclimatology of Europe from mid-winter to late spring. Climate 2: 296–309.

Please also note the supplement to this comment:

[Figure]

https://cp.copernicus.org/preprints/cp-2020-125/cp-2020-125-AC2-supplement.pdf

---

## Author Comment (AC3) · 19 Jan 2021

Thanks for your comment on my manuscript.

I agree it's going to be changed in the revised manuscript to the NGRIP record.

———————————————

---

## Author Comment (AC4) · 19 Jan 2021

The manuscript submitted to CP by Jacek Pawlak discusses an interesting multi-proxy speleothem record from Slovakia that spans MIS6/MIS6.

The new paleoclimate data are very interesting, however, the manuscript is poorly writ-

ten. The manuscript needs some deep re-structuring/re-writing. The English language style might benefit from a language editor.

The manuscript has a high potential for CP after being revised thoroughly.

My comments are list below: - Abstract:

1- something is missing before you start presenting the JS9 stalagmite. Please present the 'problematic', the questions that you're trying to answer, before talking about JS9.

This part has been rearranged.

Presently the region of central Europe is in transitional climate zone under influence of both oceanic and continental climate and continental climate. However, in the past, the region could be under stronger influence of the continental climate during cold glacial episodes or under stronger influence of oceanic climate during wetter interglacials. The long time speleothem records can adds new helpful data about past climate changes in the region. The multiproxy record of the JS9 stalagmite, collected in Demänová Cave System (Slovakia), represents ca. 60 ka period (143 – 83 ka).

2- please clarify what you mean by "transitional and continental climate"

The whole sentence has been rearranged

Presently the region of central Europe is in transitional climate zone under influence of both oceanic and continental climate and continental climate. However, in the past, the region could be under stronger influence of the continental climate during cold glacial episodes or under stronger influence of oceanic climate during wetter interglacials.

3- why do you have to mention "in opposition to the records from the Alps and the northern Tatra mountains" in the abstract? if it is so important, then please explain what you concluded about this difference with the Alps and Tatra..

In opposition to the records from the Alps and the northern Tatra mountains, the $\delta$18O record of JS9 has instant decrease episodes during Termination II. It shows that

Carpathian Belt was important climatic barrier at that time.

- Introduction:

4- replace 'most suitable" with "most commonly used"

It will be corrected in revised manuscript.

5- line 30: references are missing after "nordic seas" and "Atlantic ocean".

It has been updated

The other potential sources are the Mediterranean Sea, the Black Sea, and Nordic Seas (Ionita, 2014)

6- line 40: please explain what do you mean by stating that the speleothem 18O can be influenced by PCP? and add references to the new statement as well. As far as I know, PCP influences mostly 13C and not 18O, but I might be wrong. Please check...

You are right, It affect the $\delta$13C. It is a mistake and will be corrected in revised manuscript

7- between line 55 and 60: something is missing before you write "we present ca. 60...". Please state why a new speleothem record is needed before you present it.

It has been rearranged

However, in the past, the local climate could be more continental during colder and dryer glacial periods and more transitional at warmer interglacial periods. The new long time speleothem records can adds new helpful data about past climate changes in this region. We present ca. 60 ka long multiproxy record ($\delta$18O, $\delta$13C, Mg, Sr, Ba, Na, P, Fe, Mn, Si) of MIS-5/MIS-6 age stalagmite collected in the Demänová Cave System which is located in Slovakia.

8- before line 85: "several generations of speleothems" doesn't seem like a correct expression here.

It has been changed to:

several stages of speleothems crystallization

- Methods:

9- write "in terms of" instead of "in a term of"

It will be corrected in revised manuscript.

10- line 95: typo "oof"

It will be corrected in revised manuscript.

11-the steps described between line 95 and 100 require some re-writing

It has been rewritten to

Due to control the efficiency of chemical procedure,aAt its the beginning the spike (233U, 236U and 229Th) was added into the samples. At first step of chemical procedure, the samples were heated up for the decomposition of potential organic matter. After that the samples were dissolved in nitric acid. Finally the uranium, and the thorium were separated from the solution by chromatographic method using TRU Resin (Hellstrom, 2003).

12- line 105: "calculated by taking into account" instead of "with taking in the account"

It will be corrected in revised manuscript.

13- before line 110: "taken into account" instead of "take"

It will be corrected in revised manuscript.

14- line 110: "modified" instead of "changed"

It will be corrected in revised manuscript

15- after line 115: "minimize" instead of "minimalize"

It will be corrected in revised manuscript

-Results:

16- line 145: "described by Frisia (2015)" instead of "in the work of"

It will be corrected in revised manuscript

17- same as before (line 150).

It will be corrected in revised manuscript

18- "the" used procedure

It will be corrected in revised manuscript

19- line 160: Helltrom's procedure. A reference is needed here

The used procedure considers the possibility of contamination not only by 230Th like in original Hellstrom's procedure (Hellstrom, 2006) but also by 234U and 238U (Błaszczyk et al., 2020).

20- "relatively" slow instead of relative

It will be corrected in revised manuscript

21- between lines 170 and 175, replace "since" with "from" whenever you refer to time periods.

It will be corrected in the revised manuscript.

22- line 185: write "similar" instead of "like each other"

It will be corrected in the revised manuscript

23- replace "at that time too" with " during the same period".

It will be corrected in revised manuscript

24- general comment: refer to a figure whenever you need to mention information related to specific time periods.

It will be corrected in revised manuscript

- Discussion: 25- general comment: the main results of this paper are not highlighted in a sufficient way. The main conclusions and findings need to be well presented.

In the revised version of manuscript discussion is focused more on the topics like why the response on $\delta$18O to TII is different than in other records from Central Europe and possible explanations. The effect which is visible on all proxies around 100 ka and possible explanations.

26-replace "exemplary" with "for example" throughout the manuscript

It will be corrected in revised manuscript

27- replace "in opposition "with "contrariwise or on the opposite" throughout the manuscript

It will be corrected in revised manuscript

28- replace "the JS9 stalagmite" with "the stalagmite JS9" throughout the manuscript

It will be corrected in revised manuscript

29- replace "dryer" with "drier" throughout the manuscript

It will be corrected in revised manuscript.

Thank you, a lot, for all grammar commends they will be corrected. Additionally, the text will be sent for final language and grammar corrections. After all substantive changes and corrections.

- Conclusion: 30- I would rather write the conclusion in the form of a paragraph instead of bullet points.

[Figure]

The conclusions will be rewritten and the two topics the TII why it is so different here from other central European sites will be more highlighted.

Ionita, M., 2014 The impact of the East Atlantic/Western Russia pattern on the hydroclimatology of Europe from mid-winter to late spring. Climate 2: 296–309.

Please also note the supplement to this comment:
https://cp.copernicus.org/preprints/cp-2020-125/cp-2020-125-AC4-supplement.pdf

---

## Author Response (AR2)

**Editor commends**

*I would like to ask you to implement some technical corrections.*

Thank you for carefully reading all corrections have been applied

*Throughout the manuscript you are using either JS9 stalagmite or stalagmite JS9 or JS-9 stalagmite (and the same with record and speleothem). I don't have a recommendation for one or another but,*
*please, be coherent.*

It is corrected now JS9 is the only one form

*l.17: … a ca. 60ka period*

It is corrected

*l.37: … the Fennoscandian ice sheet (which is …*

It is corrected

*l.44: please indicate which amount you mean.*

It should be amount effect - corrected

*l.65: the reference Rybak et al. 2018 is missing.*

It is corrected

*l.86: Sotak and Borsanyi, 2004. Is it 2002 or is it a missing reference?*

It should be 2004 on reference list, checked and corrected

*l.118: 'Apart from the regular samples'. Do you mean 'In addition to the regular samples'?*

Yes it has been corrected

*l.201: from ca. -8 to 110 ka. This does not make sense to me. You can say from 130 ka to 110 ka (for example) or from -8 to -6, but I don't understand how you can travel from a ‰ to a ka.*

It is corrected
$\delta^{13}C$ values decrease from -2 to -7‰ (5‰), and the values are low ca. -8‰ until 110 ka.

*l.249. It is Cobre on the figure and Cobra in the text. Please be coherent.*

Checked and corrected

*l.267: DCS instead of DSC.*

Checked and corrected

*l.285: Drier instead of dryer.*

It is corrected

*l.286: the reference Hu et al 2005 is missing.*

It is corrected

*l.291: DCS instead of DSC.*

Checked and corrected

*l.305: The reference Shu et al. 2020 is missing.*

It is corrected

*l.315: a ca. 1.2‰ …. (or a rapid decrease of ca. 1.2‰).*

Checked and corrected

*l.322: by a ca. 2‰ … (or an instant decrease of ca. 2‰).*

Checked and corrected

*l.330: 'as a mean temperature of 1°C between the Holocene and MIS 5e'. Do you mean a mean temperature difference? Is it decrease or increase?*

Checked and clarified

According to the present temperature gradient in Slovakia, the -0.4 ‰ difference could be interpreted as a lower mean temperature of MIS 5e ca. 1C° in comparison to the Holocene. However, this simple interpretation of has a low probability.

*l.367: … a similar episode … is observed …*

It is corrected

*l.383: The reference Frisia et al. 2015 is missing.*

It should be Frisia, 2015 not Frisia et al. 2015 checked and corrected

*l.570: There is something happening with this reference when it is printed (additional or missing characters)*

It was reformatted and should work fine now.

*l.721: The reference Chapellaz et al. 1997 is missing.*

This was reference to GRIP data set it has been changed to NGRIP Data references

Andersen, K., Azuma, N. and all North Greenland Ice Core Project members 2004. High-resolution record of Northern Hemisphere climate extending into the last interglacial period. Nature 431, 147–151.

*l.721: Is the reference Pawlak et al., 2020 – submitted the same as the published one or another one ? In the latter case, it is missing.*

It is the same as Pawlak et al., 2020 checked and corrected

*l.721 and figure 6: Is it GRIP or NGRIP? The caption says GRIP while the figure indicates NGRIP.*

Checked and corrected with updated reference.

*You should include a proper 'availability section' (see author's instruction) including where your data are available (DOI: 700 10.6084/m9.figshare.13116506) [unfortunately, the data are not available for review at that address]. It would be great if the age model was available at the same place, in addition to meta-data (I cannot check if this is the case). The data availability sections should also indicate where the other data are available (where did you find them?). This is the case for all the data in figure 6.*

It was because of file embargo. The data should be available now

https://figshare.com/articles/dataset/Speleothem_oxygen_record_-_thermal_or_moisture_changes_proxy_A_case_study_of_multiproxy_record_from_MIS_5_MIS_6_age_speleothems_from_Dem_nov_Cave_System_/13116506

Other data used for comparison (fig. 6) are available at : https://www.ncdc.noaa.gov/data-access/paleoclimatology-data/datasets and in supplementary materials of cited papers in fig. 6 caption.

**Anonymous Referee #1**

Jacek Pawlak presents a new speleothem record from Slovakia that covers Termination 2 and the last Interglacial. The record is composed of multiple geochemical proxies (stable O and C isotopes, trace elements, and carbonate microfabrics) and supported by a U-Th chronology.

The record is new and of good quality, but I think the discussion of the results and placement in the regional and temporal context needs substantial more work before being accepted in CP.

I found it quite difficult to follow the discussion in several instances. Part of this might

be related to wording and language. I made some suggestions for improvement but more could be done to clarify the text. In addition, I think the figures 4 and 6 can be improved as they are quite hard to read, at the moment. I made some suggestions in the specific comments:

*I would suggest reorganizing the discussion along three main sub-headings:*

1) *Drivers of δ18O in Slovakia and comparison with other European δ18O records.*
2) *Interpretation of the other proxies.*
3) *The temporal evolution of the proxies in JS9.*

I agree that such rearranging of the discussion may improve clearness of the manuscript.

*1) Drivers of $\delta^{18}O$ in Slovakia and comparison with other European $\delta^{18}O$ records.*

*Here, I found it difficult to understand what is meant by "thermal" control, as $\delta^{18}O$ can be influenced by temperature in different ways, which the author describes, but then does not further elaborate on. More clarity in the language (what is meant by "thermal" or "temperature" effect in $\delta^{18}O$ at the different locations described?)*

*and a more in-depth discussion of the likely controls on $\delta^{18}O$ at the DCS cave is needed. For example, is there monitoring data from the cave that can back up some of the interpretation of $\delta^{18}O$? Nearby GNIP stations that can be used to test the modern relationship between temperature, moisture source, and $\delta^{18}O$ in precipitation? So far, this part of the discussion is tenuous and seems based mostly on speculation.*

I agree that this dependence is complicated. I believe that it is more the local thermal signal (changes in mean annual temperature) which influences the $\delta^{18}O$ of precipitation and has impact on isotopic fractionation during calcite crystallization in the cave. However, in longer time scale the isotopic composition of ocean source which depend on mean global temperature and global volume of glaciers becomes important (source effect). Additionally, the circulation changes which causes the changes in proportion of vapors from different sources plays important role here.

There are several stations in Slovakia, where the $\delta^{18}O$ of precipitation was measured continuously (Holko et al 2012). The detailed study in the region shows, that there is a strong dependence between the $\delta^{18}O$ and temperature. This dependence is observed in the long time monitoring for single sites (Holko et al 2012). Exemplary, for the closest station next to the research area in Liptovski Mikulas the $R^2$ of correlation between the $\delta^{18}O$ of atmospheric precipitation and temperature is 0.639 (Holko et al 2012). Additionally, there is a stronger dependence between mean annual temperature of the site, and mean $\delta^{18}O$ value in Slovakia with $R^2 = 0.728$ (Holko et al 2012). The dependence between mean annual precipitation and mean $\delta^{18}O$ value is less visible ($R^2 = 0.483$) (Holko et al 2012).

There is other methodological work which includes data from many cave sites located on several continents the conclusion of this work is that sites with mean annual

temperature lower than 15C and the aridity index higher that 0.65 has a potential for growing speleothems which reflects the mixed signal of past temperature and past precipitation ( Baker et al 2018).

All this information will be included in revised version of discussion.

*2)* ***Interpretation of the other proxies****. Again, I would welcome some more detailed reasoning on why the proxies are interpreted in the way they are. For $\delta^{13}C$, it would be useful to know more about the soil cover and vegetation assemblage above the cave. Are there palynological/palaeoecological studies from the region that could shed light on expected changes in biosphere responses over glacial-interglacial timescales? The discussion of the trace elements similarly lacks depth. It is not clear to me whether the author is implying a control of prior calcite precipitation (PCP) control on Mg, Sr, and Ba, or whether the dominant control is thought to be the dolomite dissolution. In any case, this should be elaborated: what is the basis for the claim that dolomite dissolution is dominant? Are there measurements of the host rock composition? Overall, I think there is currently some overinterpretation of small wiggles in the trace element records (Mg, Sr, Ba), in particular. I don't see much variability in the older part of the Mg record. I would also caution against interpreting the trace element record at the base of the stalgmite, as this is often a region where effects that have nothing to do with climate play a role.*

Presently the vegetation cover over the Demianovska Cave System is dominated by the mixed forest of mountain type and grasslands connected with mountain slopes activity. Therefore, presently there are two types of soils, over the cave, associated with forest and the other type associated with slopes activity (Hercman et al 2020).

There are few palynological sites nearby. The important one is located in a closed proximity is Safarka (Jankovska, 2002). The pollen record from Safarka covers ca. last 60ka and recording changes of plants cover during the last glacial. The record show a predominance of needle-leaved and cold temperate tree vegetation during warmer episodes of MIS 3. The Last Glacial Maximum part of these pollen record suggests that some woody cover was maintained in the region during the maximum northern hemisphere ice extent (between 26.5 and 19 ka cal BP). The beginning of Holocene is marked by introduction of temperate forest (Feurdean et al 2014). The changes in vegetation at the end of last glacial and the begging of Holocene are well expressed in $\delta^{13}C$ records from Demianova Cave System (Hercman et al 2020).

The Demianova Cave is developed mostly in Gutenstein limestones (Early Anisian) and  Ramsau dolomite (Ladinian) therefore both the dissolution of limestones and dolomites can happened. During periods of longer water residence time the contribution of Magnesium from Dolomite source is increased. This effect results in higher Mg/Ca values and lover Sr/Ca values (Tremaine and Froelich 2013). The PCP also can occur during dry episodes of longer water residence times. However, the PCP results in increase of all X/Ca rations (Tremaine and Froelich 2013) which is rather not observed in a studied record.

*3) **The temporal evolution of the proxies in JS9**. In my opinion, the records are overinterpreted at this point. Many of the smaller wiggles are probably not resolvable given the chronological uncertainty in the record. I would suggest the author focus and expand the discussion on the larger and interesting features in the record, e.g., the large peak in $\delta^{13}C$, P, Fe, and Mn around 100 ka BP, as well as why TII is only weakly expressed in the record. It is possible that with some restructuring of the text this is already possible, but as mentioned above I found it quite difficult to follow at the moment. I think it would be better to use a more recent ice core record (NGRIP for example) to compare the speleothem records to.*

I agree with your commend about overinterpretation of some part of that record in revised version of the Manuscript the discussion will be oriented more on the most important fact like the large event around 100 ka visible in this record and the expression of TII.

In previous version I decided to use GRIP due to the fact, that it covers whole studied period. However, after I got several commends and suggestions about that I agree that the NGRIP is better solution even if it covers only part of the studied record.

***Specific comments: - line 27 and rest of text: use "precipitation" instead of "rain precipitation"***

It will be corrected in revised manuscript.

*- **line 44 and following: I would not include Middle Eastern records in the discussion about European records. Rather discuss the European records and then add a sentence. showing the similarities with the Middle East and linking that to the prevailing circulation patterns?***

It has been changed.

In contrast to the most of European records, the records from Middle-East seems to be influenced by more factors, like the amount of precipitation, temperature, and also changes in the main source of vapor for rain precipitation (source effect; Bar-Matthews et al., 2003). It can be linked to changes in the prevailing circulation patterns, the impact of evaporation on Mediterranean Sea surface $\delta^{18}O$ and also lower amplitude of long time mean annual temperature changes during Last Interglacial at lower latitudes (Rybak et al. 2018).

*- **line 52: I don't understand this sentence.***

It has been changed to

Presently, Slovakia is influenced by two main types of climates (Kottek et. al. 2006), the boreal fully humid with warm summers climate (Dfb) on the East and warm temperate fully humid climate (Cfb) on the West.

*- line 55: Since it's a single author paper, change the "we" to first person.*

It will be corrected in revised manuscript.

*- line 67: I think it should be "genesis of DCS" instead of "genese"*

It will be corrected in revised manuscript.

*- line 78 and in other parts of the text: "peak" instead of "pick"*

It will be corrected in revised manuscript.

*- line 95: Why were samples for dating drilled to be as thick as possible? This seems counterintuitive, as one would typically try and minimize the amount of sample to avoid integrating too much time within a single sample.*

It is a mistake it should be thin here.

*- line 117: "To minimalize the difference in resolution between the lower and upper part of the studied record caused by the sedimentation rate, which is slower for the lower part. The lower part of a stalagmite from 0 to 40 mm was additionally sampled with a resolution of one sample/0.3 mm." I think these two sentences should be joined, as the first one does not make sense on its own.*

The lower part of a stalagmite ( 0 to 40 mm) was additionally sampled with a resolution of one sample/0.3 mm, to minimalize the difference in resolution between the lower and upper part of the studied record caused by, slower for the lower part sedimentation rate.

*- line 118: Use "growth rate" instead of "sedimentation rate" for speleothems*

It will be corrected in revised manuscript.

*- line 212 and following: instead of "short time signal" it would be clearer to refer to "short term variability".*

Thank you, a lot, for all grammar commends they will be corrected. Additionally, the text will be sent for final language and grammar corrections. After all substantive changes and corrections.

*- line 233: I think it's interesting that the TII is only visible as such a muted response in $d^{18}O$, compared to the overall variability in the record. I would be interested in knowing more about why that is.*

I agree. Additional question is why the final negative response of $\delta^{18}O$ at TII is opposite to response observed in other Central European sites exemplary in northern part of Tatra Montains (Magurska Cave) and in Alpine records (Holloch and Schneckenloch). The response in those Central European sites is clearly connected,

with two processes first with increase of mean annual temperature and second with change from winter dominated precipitation to summer dominated precipitation (Moseley et al. 2015; Meyer et al., 2008; Holzkamper et al., 2004). In case of studied site, the observed final negative response must be caused by local or regional effect which was stronger that thermal effect at that time. The possible effect which my cause the lower value of $\delta^{18}O$ is circulation effect and change from sources of precipitation like Adriatic Sea or Black Sea to Atlantic Source and vapor recycled over the European continent. The instant change to more negative values may be caused by source and continental effect, which overcame the temperature effect at that time. Therefore, the muted response is a result of both the effect of increase the mean annual temperature and source/circulation effect which overcomes each other.

*- line 255: The growth rate appears to be much lower in the interval 127-122, which stands at odds with the interpretation of the isotopes (wetter climate and well-developed soil). Any thoughts on why that might be? Also, I don't agree with the following sentence on Mg, as I don't think there is a significant trend there.*

The interval of low growth rate is much longer it starts at ca. 133 ka and ends at ca. 112. In my opinion it may be caused by local effects. Our research of five Holocene stalagmites from Demianova Cave System shows that changes in growth rate are different for all studied stalagmites and they are not connected directly with climate changes (Hercman et al., 2020). Therefore, they must be caused by local effect in the cave.

I agree with your opinion that trace elements like Mg must be treated with caution. It is true that the Mg record is dominated rather by pseudo cyclic changes without the significant main trend. However, there are two short intervals where the Mg content is significantly lower at ca. 100 and 88 ka, which is interesting. Additionally, in my opinion there are several intervals of lower Mg values which repeats $\delta^{13}C$ signal at 88, 100, 110, 113 and 124 ka. In my opinion, it is an argument that those intervals may relate to increased amount of precipitation.

*- Figure 1: the map of Slovakia could be improved by showing the (Tatra Mountains, Low Tatra moununtains, location of Magurska Cave and the boundaries of climate zones or dominant air masses.*

[Figure]

The boundaries of climate zones, locations of Tatra Mts and Low Tatra Mts and the caves in the nearest proximity (Magurska and Baradla ) have been added to fig 1A.

*- Figures 4 and 6: I find these two figures very hard to read, as there is a lot of information. Given that they are so crucial to the discussion, I have a few suggestions to improve them: 1. I would add a second age scale at the top of the figures, to make it easier to read 2. Clearly label the TII, maybe as a bar that covers the figure 3. I wonder if it is necessary to show all the records in figure 6, I think focusing on some key records from each group (maybe the longest ones?) would make it easier to read. But this might be my personal preference.*

[Figure]

[Figure]

According to your suggestion I added the additional age scale on the top of both figures and solid line for TII and dotted lines between MIS 5e/5d/5c/5b boundaries.

*I like the color scheme linking back to figure 5, please add the explanation of the different groups to the caption.*

Thank you. The explanation of the color scheme has been added to the caption bellow fig 6.

Fig. 6. The comparison of JS9 $\delta^{18}$O record with other records of MIS-5/MIS-6 age from Europe and Middle East. GRIP (Chappellaz *et al.,*1997); Magurska (Pawlak et al., 2020 – *submitted*); Baradla (Demény et al., 2017); Orlova Tchuka (Pawlak et al., 2019); Schneckenloch (Mosley *et al.,* 2015); Holloch (Moseley *et al.,* 2015); Entrische Kirche (Meyer *et al.,* 2008); Bourgeois-Delaunay (Couchoud et al. 2009); Cobre (Rossi et al. 2014); Han-sur-Lesse (Vansteenberge et al., 2016); Antro del Corchia (Drysdale *et al.,* 2005); Soreq (Bar-Matthews *et al.,* 2003); Peqiin (Bar-Matthews *et al.,* 2003); Kanaan (Nehme *et al.,* 2015); black and gray colors charts - speleothems with temperature as a dominant factor influencing on $\delta^{18}$O value; Green colors charts – speleothems, where the changes of the isotopic composition of rainwater and amount of precipitation are dominant factors influencing on $\delta^{18}$O value; Red color charts - studied site.

Baker, A., Hartmann, A., Duan, W. *et al.* 2019. Global analysis reveals climatic controls on the oxygen isotope composition of cave drip water. Nature Communications 10, 2984. https://doi.org/10.1038/s41467-019-11027-w

A. Feurdean, A. Perşoiu, I. Tanţău, T. Stevens, E.K. Magyari, B.P. Onac, S. Marković, M. Andrič, S. Connor, S. Fărcaş, M. Gałka, T. Gaudeny, W. Hoek, P. Kolaczek, P. Kuneš, M. Lamentowicz, E. Marinova, D.J. Michczyńska, I. Perşoiu, M. Płóciennik, M. Słowiński, M. Stancikaite, P. Sumegi, A. Svensson, T. Tămaş, A. Timar, S. Tonkov, M. Toth, S. Veski, K.J. Willis, V. Zernitskaya, 2014. Climate variability and associated vegetation response throughout Central and Eastern Europe (CEE) between 60 and 8 ka, Quaternary Science Reviews;106, 206-224,

Hercman H, Gąsiorowski M, Pawlak J, et al. 2020. Atmospheric circulation and the differentiation of precipitation sources during the Holocene inferred from five stalagmite records from Demänová Cave System (Central Europe). The Holocene. 30(6):834-846. doi:10.1177/0959683620902224

Holko, L., Dóša, M., Michalko, J., & Šanda, M. 2012. Isotopes of oxygen-18 and deuterium in precipitation in Slovakia / Izotopy kyslíka-18 A deutéria v zrážkach na Slovensku, Journal of Hydrology and Hydromechanics, *60*(4), 265-276. doi: https://doi.org/10.2478/v10098-012-0023-2

Jankovska, V., Chromý, P., Niznianska, M., 2002. Safarka - first palaeobotanical data on vegetation and landscape character of Upper Pleistocene in West Carpathians (North East Slovakia). Acta Palaeobotanica 42, 29-52.

Tremaine D M, Froelich P N, 2013. Speleothem trace element signatures: A hydrologic geochemical study of modern cave drip waters and farmed calcite, Geochimica et Cosmochimica Acta 121; 522-545.

*Anonymous Referee #2*

*The manuscript submitted to CP by Jacek Pawlak discusses an interesting multi-proxy speleothem record from Slovakia that spans MIS6/MIS6.*

*The new paleoclimate data are very interesting, however, the manuscript is poorly written. The manuscript needs some deep re-structuring/re-writing. The English language style might benefit from a language editor.*

*The manuscript has a high potential for CP after being revised thoroughly.*

*My comments are list below:*

*- Abstract:*

*1- something is missing before you start presenting the JS9 stalagmite.*
*Please present the 'problematic', the questions that you're trying to answer, before talking about JS9.*

This part has been rearranged.

Presently the region of central Europe is in transitional climate zone  under influence of both oceanic and continental climate and continental climate. However, in the past, the region could be under stronger influence of the continental climate during cold glacial episodes or under stronger influence of oceanic climate during wetter interglacials. The long time speleothem records can adds new helpful data about past climate changes in the region. The multiproxy record of the JS9 stalagmite, collected in Demänová Cave System (Slovakia), represents ca. 60 ka period (143 – 83 ka).

*2- please clarify what you mean by "transitional and continental climate"*

The whole sentence has been rearranged

Presently the region of central Europe is in transitional climate zone under influence of both oceanic and continental climate and continental climate. However, in the past, the region could be under stronger influence of the continental climate during cold glacial episodes or under stronger influence of oceanic climate during wetter interglacials.

*3- why do you have to mention "in opposition to the records from the Alps and the northern Tatra mountains" in the abstract? if it is so important, then please explain what you concluded about this difference with the Alps and Tatra..*

In opposition to the records from the Alps and the northern Tatra mountains, the $\delta^{18}O$ record of JS9 has instant decrease episodes during Termination II. It shows that Carpathian Belt was important climatic barrier at that time.

*- Introduction:*

*4- replace 'most suitable" with "most commonly used"*

It will be corrected in revised manuscript.

*5- line 30: references are missing after "nordic seas" and "Atlantic ocean".*

It has been updated

The other potential sources are the Mediterranean Sea, the Black Sea, and Nordic Seas (Ionita, 2014)

*6- line 40: please explain what do you mean by stating that the speleothem 18O can be influenced by PCP? and add references to the new statement as well. As far as I know, PCP influences mostly 13C and not 18O, but I might be wrong. Please check...*

You are right, It affect the $\delta^{13}$C. It is a mistake and will be corrected in revised manuscript

*7- between line 55 and 60: something is missing before you write "we present ca. 60...". Please state why a new speleothem record is needed before you present it.*

It has been rearranged

However, in the past, the local climate could be more continental during colder and dryer glacial periods and more transitional at warmer interglacial periods. The new long time speleothem records can adds new helpful data about past climate changes in this region. We present ca. 60 ka long multiproxy record ($\delta^{18}$O, $\delta^{13}$C, Mg, Sr, Ba, Na, P, Fe, Mn, Si) of MIS-5/MIS-6 age stalagmite collected in the Demänová Cave System which is located in Slovakia.

*8- before line 85: "several generations of speleothems" doesn't seem like a correct expression here.*

It has been changed to:

several stages of speleothems crystallization

*- Methods:*

*9- write "in terms of" instead of "in a term of"*

It will be corrected in revised manuscript.

*10- line 95: typo "oof"*

It will be corrected in revised manuscript.

*11-the steps described between line 95 and 100 require some re-writing*

It has been rewritten to

Due to control the efficiency of chemical procedure,aAt its the beginning the spike ($^{233}$U, $^{236}$U and $^{229}$Th) was added into the samples. At first step of chemical procedure, the samples were heated up for the decomposition of potential organic matter. After that the samples were dissolved in nitric acid. Finally the uranium, and the thorium were separated from the solution by chromatographic method using TRU Resin (Hellstrom, 2003).

*12- line 105: "calculated by taking into account" instead of "with taking in the account"*

It will be corrected in revised manuscript.

*13- before line 110: "taken into account" instead of "take"*

It will be corrected in revised manuscript.

*14- line 110: "modified" instead of "changed"*

It will be corrected in revised manuscript

*15- after line 115: "minimize" instead of "minimalize"*

It will be corrected in revised manuscript

***-Results:***

*16- line 145: "described by Frisia (2015)" instead of "in the work of"*

It will be corrected in revised manuscript

*17- same as before (line 150).*

It will be corrected in revised manuscript

*18- "the" used procedure*

It will be corrected in revised manuscript

*19- line 160: Helltrom's procedure. A reference is needed here*

The used procedure considers the possibility of contamination not only by $^{230}$Th like in original Hellstrom's procedure (Hellstrom, 2006) but also by $^{234}$U and $^{238}$U (Błaszczyk et al., 2020).

*20- "relatively" slow instead of relative*

It will be corrected in revised manuscript

*21- between lines 170 and 175, replace "since" with "from" whenever you refer to time periods.*

It will be corrected in revised manuscript.

*22- line 185: write "similar" instead of "like each other"*

It will be corrected in revised manuscript

*23- replace "at that time too" with " during the same period".*

It will be corrected in revised manuscript

*24- general comment: refer to a figure whenever you need to mention information related to specific time periods.*

It will be corrected in revised manuscript

*- Discussion:*

*25- general comment: the main results of this paper are not highlighted in a sufficient way. The main conclusions and findings need to be well presented.*

In the revised version of manuscript discussion is focused more on the topics like why the response on $\delta^{18}$O to TII is different than in other records from Central Europe and possible explanations. The effect which is visible on all proxies around 100 ka and possible explanations.

*26-replace "exemplary" with "for example" throughout the manuscript*

It will be corrected in revised manuscript

*27- replace "in opposition "with "contrariwise or on the opposite" throughout the manuscript*

It will be corrected in revised manuscript

*28- replace "the JS9 stalagmite" with "the stalagmite JS9" throughout the manuscript*

It will be corrected in revised manuscript

*29- replace "dryer" with "drier" throughout the manuscript*

It will be corrected in revised manuscript.

Thank you, a lot, for all grammar commends they will be corrected. Additionally, the text will be sent for final language and grammar corrections. After all substantive changes and corrections.

*- Conclusion: 30- I would rather write the conclusion in the form of a paragraph instead of bullet points.*

The conclusions will be rewritten and the two topics the TII why it is so different here from other central European sites will be more highlighted.

Ionita, M., 2014 The impact of the East Atlantic/Western Russia pattern on the hydroclimatology of Europe from mid-winter to late spring. Climate 2: 296–309.

---

## Author Response (AR3)

*line 45: I'm not sure I understand why the PCP influence was deleted here. Is it because it won't strongly affect d18O? In that case, please specify that this sentence refers to d18O (instead of "isotopic composition"). If not, please clarify.*

I think the best way is add it again with additional sentence
*"The isotopic composition of dripping waters can also be modified inside the soil and in the epikarst zone by evaporation and priori calcite precipitation (PCP; Baker et al., 2019). However, PCP affects more the $\delta^{13}C$ value."*

*line 51: I would again remove the references for records from Lebanon and Israel here since they do not technically belong to Europe.*

Those two references have been removed from paragraph about European records.

*line 67: "However, in the past, the local climate was more continental during colder and drier glacial periods and more transitional at warmer interglacial periods" this sentence needs a reference.*

It has been fixed.

*line 230: "The $\delta18O$ records in the regions, where the aridity index is lower than 0.65 reflects stronger influence of evaporation." This sentence is irrelevant here, please delete.*

This sentence has been removed.

*line 230 and following: I was wondering if the more arid climate of the glacial could really result in evaporation, if temperatures are also lower. Maybe the author could indicate whether temperature or aridity would likely be the dominant factor over glacial-interglacial cycles.*

The main factor in the long time scale is a temperature. However, in short time scale the influence of humidity/aridity can affect the main temperature signal. I agree that at the low temperature conditions the evaporation is low. However, even during episodes of glacial conditions the DSC located at ca. 48 N latitude was affected by quite high insolation, especially during the continental summer. However, it is impossible to tell basing on data presented here, how the real impact could be. I modified the end of this paragraph.

I put the sentence about the importance of temperature and aridity in case of DSC at the end of the whole paragraph

*line 242: I think a statement that links back to the DCS is needed after the discussion of the other European records. It should highlight again what the temperature signal of d18O in DCS speleothems means (temperature effect on precipitation isotopes). Also highlight the temperature influence on calcite precipitation (-0.18 permil/°C after Tremaine et al., 2011) and how that compares to the precipitation gradient (0.36 permil/°C). The net effect is therefore smaller than the precipitation effect.*

I add the summarizing stamen at the end of this paragraph.

"*According to all the facts listed above it can be assumed that, in the long time scale, the temperature effect on atmospheric precipitation should be the main factor shaping the $\delta^{18}O$ value of the DSC stalagmites. The whole temperature effect on $\delta^{18}O$ value of speleothem calcite is lover than presently observed $\delta^{18}O$ temperature gradient of precipitation (0.36 ‰/°C; Holko et al 2012) due to the opposite temperature effect on calcite crystallisation (-0.18 ‰/°C). Therefore, the expected net effect should be ca 0.18 ‰/°C. However, in the short time scale the main temperature effect can be changed by humidity/aridity effects like it was described in case of Bourgeois–Delaunay Cave (Couchoud et al., 2009).'*

"

*line 245: Carbon is also sourced from the atmosphere.*

That is true. I modified that sentence *" CO2 from a soil source can reflect the changes in the isotopic composition of atmospheric CO2 and additionally is enriched in 12C due to biological activity. Due to that fact a well-developed soil cover results in a lower $\delta13C$ value"*

*line 284 and following: I wonder if there is a chance that the anomalous d13C and trace element values are related to the presence of micrite and reflecting anomalous growth at the base of the stalagmite. I would be cautious with the climatic interpretation of the base of the stalagmite.*

It is important topic and in my opinion, it may be possible explanation for the episode around 105 ka, when all listed proxies react in the same way. The situation at the base of stalagmite is different the oxygen react in a different way than carbon. However this very short pick directly around 140 ka is similar and can be raised by similar effect I added the sentence about that.

*line 337: Please again add a statement that clarifies how the DCS record is interpreted in this period. What are the possible explanations for the discrepancy between global (warm) and regional (cool) climate conditions?*

It has been clarified at the end of that paragraph.